# Long-term profiling of mineral dust and pollution aerosol with multiwavelength polarization/Raman lidar at the Central Asian site of Dushanbe, Tajikistan: Case studies

Julian Hofer[1], Dietrich Althausen[1], Sabur F. Abdullaev[2], Abduvosit N. Makhmudov[2], Bakhron I. Nazarov[2], Georg Schettler[3], Ronny Engelmann[1], Holger Baars[1], K. Wadinga Fomba[1], Konrad Müller[1], Bernd Heinold[1], Konrad Kandler[4], and Albert Ansmann[1]

[1]Leibniz Institute for Tropospheric Research, Leipzig, Germany
[2]Physical Technical Institute of the Academy of Sciences of Tajikistan, Dushanbe, Tajikistan
[3]Helmholtz Center Potsdam, German Research Center for Geosciences, Potsdam, Germany
[4]Institut für Angewandte Geowissenschaften, Technische Universität Darmstadt, Darmstadt, Germany

*Correspondence to:* Julian Hofer, hofer@tropos.de

**Abstract.** For the first time, continuous vertically resolved aerosol measurements were performed by lidar in Tajikistan, Central Asia. Observations with the multiwavelength polarization/Raman lidar Polly$^{XT}$ were conducted during CADEX (Central Asian Dust EXperiment) in Dushanbe, Tajikistan, from March 2015 to August 2016. Co-located with the lidar a sun photometer was operated. The goal of CADEX is to provide an unprecedented data set on vertically resolved aerosol optical properties in Central Asia, an area highly affected by climate change but largely missing vertically resolved aerosol measurements. During the 18-months measurement campaign, mineral dust was detected frequently from ground to cirrus level height. In this study, an overview of the measurement period is given and four typical but different example measurement cases are discussed in detail. Three of them are dust cases and one is a contrasting pollution aerosol case. Vertical profiles of the measured optical properties and the calculated dust and non-dust mass concentrations are presented. Dust source regions were identified by means of backward trajectory analyses. A lofted layer of Middle Eastern dust with an aerosol optical thickness (AOT) of 0.4 and an extinction-related Ångström exponent of 0.41 was measured. In comparison, two near-ground dust cases have Central Asian sources. One is an extreme dust event with an AOT of 1.5 and Ångström exponent of 0.12 and the other one is a most extreme dust event with an AOT of above 4 (measured by sun photometer) and an Ångström exponent of -0.08. The observed lidar ratios (particle linear depolarization ratios) in the presented dust cases range from 40.3 sr to 46.9 sr (0.18–0.29) at 355 nm and from 35.7 sr to 42.9 sr (0.31–0.35) at 532 nm wavelength. The particle linear depolarization ratios indicate almost unpolluted dust in the case of a lofted dust layer and pure dust in the near-ground dust cases. The lidar ratio values are lower than typical lidar ratio values for Saharan dust (50–60 sr) and comparable to Middle Eastern/West-Asian dust lidar ratios (35–45 sr). In contrast, the presented case of pollution aerosol of local origin has an Ångström exponent of 2.07 and a lidar ratio (particle linear depolarization ratio) of 55.8 sr (0.03) at 355 nm and 32.8 sr (0.08) at 532 nm wavelength.

# 1 Introduction

Atmospheric mineral dust can be transported over tens of thousands of kilometers away from its sources, arid and semi-arid source regions (Uno et al., 2009; Haarig et al., 2017). More observations, especially in western and Central Asia, are needed to describe global and regional dust transport and to estimate the effect of this dust on air quality (Chin et al., 2007), and climate, via direct and various indirect radiative effects. Extended investigations to Saharan dust close to its source regions (e.g. SAMUM–1,2 (Saharan Mineral Dust Experiment), Fennec climate programm, SHADOW (Study of SaHAran Dust Over West Africa)) (Heintzenberg, 2009; Ansmann et al., 2011a; Ryder et al., 2015; Veselovskii et al., 2016) as well as regarding dust long range transport across the Atlantic ocean (e.g. SALTRACE (Saharan Aerosol Long-range Transport and Aerosol-Cloud-Interaction Experiment)) (Weinzierl et al., 2017) have been conducted to obtain novel data to reduce uncertainties in above mentioned estimations.

However, the global dust belt, which reaches from the Sahara over the Arabian deserts to the Taklamakan and Gobi deserts, contains a lot more arid and semi-arid regions which act as sources for atmospheric mineral dust in the northern hemisphere (Darmenova et al., 2009; Ridley et al., 2016). This dust has a sensitive impact on climate, environmental conditions, ecosystems, and health. Central Asia lies in the middle of this global dust belt and contains some major dust sources (Fig. 1). Therefore, Central Asian countries are frequently affected by atmospheric mineral dust hazardous to respiratory health (Wiggs et al., 2003). Even bacteria, fungi, and viruses can be transported on dust on long distances (Griffin, 2007; Hara and Zhang, 2012; Yamaguchi et al., 2015; Park et al., 2016). Dust is also transported across the highly polluted Central Asia (Balance and Pant, 2003) further eastwards (Tanaka et al., 2005; Mikami et al., 2006) and on its way it is subject to stronger anthropogenic influence compared to the westward transport of Saharan dust (Bi et al., 2016).

Dust from different sources have different mineralogical compositions (Caquineau et al., 2002), different optical properties (Sokolik et al., 1993; Su and Toon, 2011; Schuster et al., 2012; Mamouri et al., 2013), and therefore different impact on radiation and climate (Sokolik and Golitsyn, 1993; Golitsyn et al., 1993b; Nazarov et al., 2010a, b). Central Asia in particular is vulnerable to climate change and is already affected by it (Freedman and Neuzil, 2015; Lioubimtseva and Henebry, 2009; Xu et al., 2016). For example, dramatic glacier shrinking took place in the last decades in Tajikistan (Kayumov, 2010; Chatenoux, 2011), which has an effect on the water resources of Tajikistan and the whole Central Asian area (Chen et al., 2017). The links between mineral dust occurrence and climate change as well as possible feedbacks have to be quantified. Therefore, aerosol profiling in Central Asia is strongly required to better understand regional and global transport and deposition of mineral dust and its effects on radiation budget, cloud and precipitation development, and human health.

The sources and emission of dust in Central Asia, interaction with aerosol pollution, and climate relevant processes are investigated with models and satellite remote sensing (Xi and Sokolik, 2016, 2015a, b; Nobakht et al., 2015). Measurements with sun photometers, despite being rare in inland Central Asia (Klüser et al., 2015; Bayat et al., 2011), can provide column integrated aerosol properties (Semenov et al., 2005; Bi et al., 2016; Chen et al., 2013b), but additionally vertically resolved information on aerosol optical properties are needed to thoroughly investigate radiative effects and dust-cloud-dynamics interaction. The widely used satellite-borne lidar CALIPSO/CALIOP (Cloud-Aerosol Lidar and Infrared Pathfinder Satellite

Observations / Cloud-Aerosol Lidar with Orthogonal Polarization) delivers snapshot-like observations in terms of backscatter profiles at 532 nm and 1064 nm wavelength. To estimate extinction coefficients of dust, the dust lidar ratio is required as an input, which introduces considerable uncertainty (e.g. Wandinger et al., 2010; Tesche et al., 2013; Amiridis et al., 2013). To obtain sophisticated sets of optical properties and detailed information on particle mixture and layering, long-range dust trans-
port, multi-day coherent structures, vertical mixing between the planetary boundary layer (PBL) and lofted aerosol layers, and aerosol-cloud interaction, continuous observations with ground-based Raman lidars or HSRL (High-Spectral-Resolution lidar) are needed. Up to now, only few experiments were performed to characterize the aerosol over Central Asia. Already in 1989 a Soviet-American research team conducted a joint experiment on dust in Tajikistan (Golitsyn and Gillette, 1993; Golitsyn et al., 1993a). Coordinated ground-based, aircraft and satellite measurements took place during two dust storms on 16 and 20
September 1989. The total area occupied by the atmospheric dust during the latter dust storm was approximately $10^5$ km$^2$ (Smirnov et al., 1993). The mass of dust in the air was estimated as 3.1 t (Smirnov et al., 1994). Chemical analysis of the collected dust (Gomes and Gillette, 1993) showed a low iron content for Central Asian dust. Also, a calcareous character and high contents of soluble salts were observed (Miller-Schulze et al., 2011; Schettler et al., 2014; Groll et al., 2013; Andronova et al., 1993). In general, however, a very high variability (Andronova et al., 1993) lead to considerably different results de-
pending on sampling location. Analysis of quartz filter samples collected in Dushanbe showed, besides significant amounts of dust, strong contribution of anthropogenic black carbon pollution from fuel combustion and smoke (Hansen et al., 1993). Optical and radiative properties have been studied as well (Shukurov et al., 1993). Sokolik and Golitsyn (1993) performed radiative transfer calculations for the dust storms in 1989 and state that the vertical distribution of additional total radiative heating rates can only be calculated with assumptions on vertically homogeneous optical properties and on the height of the
dust layer. Sokolik et al. (1993) compared complex refractive indices from the dust collected in Tajikistan with dust from other sources. They found a seemingly different spectral behavior of the complex refractive index of Central Asian dust compared to Saharan dust, but Sokolik et al. (1993) concluded that there are large uncertainties and the discrepancies could be due to different measurement techniques. The measured AOTs during those dust storms were up to 1.3 (Golitsyn and Gillette, 1993) and 3.3 at 550 nm (Pachenko et al., 1993), respectively. A few vertical profiles of the scattering coefficient of the dust were
measured with a nephelometer on board an aircraft (Pachenko et al., 1993). These measurements showed that the majority of the dust was concentrated in the lowermost 3–3.5 km below a thermal inversion. Pachenko et al. (1993) mentioned dust plumes reaching through this inversion up to 6 km altitude. The measurements on 16 September 1989 also revealed a dust layer at a height of about 4 km. Fraser (1993) used satellite data to obtain the horizontal distribution of the dust in Central Asia. Fraser (1993) emphasized the need of vertical profiles of the dust mass, which can be estimated from dust extinction coefficient, to
determine the movement of the dust. Gillette et al. (1993) modeled dust deposition and transport during this 1989 campaign and mentioned the sensitivity of their model to the dust layer height, especially because of the complicated topography in Tajikistan.

Since recently, a lidar station is operated in Eastern Kyrgyzstan (Chen and Sverdlik, 2007; Chen et al., 2013a). For this station in Kyrgyzstan, Chen et al. (2013b) report dust transported from the Aral Sea region, long-range transported dust from the
Middle East as well as dust from the Taklamakan desert. A lidar is operated in Aksu (northwestern Taklamakan) (Kai et al.,

2008), where local dust was measured from ground to a height of 6 km during strong dust outbreaks (Tsunematsu et al., 2005) as well as lofted long-range transported dust layers at heights of 11 km (Mikami et al., 2006). Lidar measurements along the Caspian and Aral Sea and the Lake Balkhash in Kazakhstan were performed by Dieudonné et al. (2015). Dieudonné et al. (2015) showed that dust layers originating in the Caspian and Aral Sea region can regularly spread over wide areas of Russia and last for several days.

Nevertheless, the knowledge of the vertical aerosol distribution over Tajikistan and especially the transport of mineral dust over Central Asia is still insufficient. Therefore, the Central Asian Dust EXperiment (CADEX) was proposed. CADEX provides long-term data on vertical profiles of particle optical properties above Tajikistan. A multiwavelength polarization/Raman lidar (Engelmann et al., 2016) was run continuously in Tajikistan over the 18-months period from March 2015 to August 2016. These measurements are part of the PollyNET (Baars et al., 2016), which is an attempt to contribute to the establishment of a ground-based polarization/Raman lidar network (e.g. Baars, 2011; Kanitz et al., 2013; Giannakaki et al., 2016), with the aim to support regional and global dust modeling.

This article provides an overview of the observations and focuses mainly on four exemplary measurement cases which show typical but different scenarios of atmospheric mineral dust in Tajikistan. The vertical profiles of the measured particle optical properties, calculations of the dust mass concentration, and backward trajectories for the individual cases are presented. In a follow-up publication, a summary of the key findings of the CADEX campaign including a statistical analysis of the entire measurement period will be provided.

In section 2 of this article, the CADEX campaign, the measurement site, the used lidar system, and the used auxiliary data are described. Section 3 gives an overview of the observations and the four exemplary measurement cases are presented in detail. The cases are compared and discussed in section 4.

## 2  The Central Asian Dust EXperiment (CADEX) and instruments

CADEX was a joint German-Tajik project to investigate the atmospheric mineral dust over Tajikistan. The goal of the project was to provide a not yet available data set of optical and microphysical properties of Central Asian mineral dust on a longer-term basis. The data are used to characterize and investigate the particle types, possible aerosol sources, and the aerosol influence on the radiation budget in Central Asia.

Tajikistan lies in the global dust belt in close proximity of some major dust sources like the Taklamakan desert, the Aralkum desert of the desiccating Aral Sea, the Kyzylkum and Karakum deserts east of the Caspian Sea, the Iranian Dasht-e-Kavir and Dasht-e-Lut deserts and the deserts in Afghanistan (Fig. 1). Therefore, Tajikistan is frequently affected by severe dust events and is a net accumulator of dust (Gillette and Dobrowolski, 1993). Tajikistan is a country with a dry continental climate and benefits from its water resources. Melt water discharge from the high mountain regions ensures a high freshwater availability. Furthermore, Central Asia and especially Tajikistan are highly affected by climate change. For example, dramatic glacier shrinking took place in the last decades (Makhmadaliev and Novikov, 2002; Makhmadaliev et al., 2004, 2008; Kayumov and Novikov, 2014). This could have also an effect on the water resources of Tajikistan and the whole Central Asian area in

the future, since the trans-regionally important rivers as Amu Darya and Syr Darya are fed by glacier and snow melt water (Siegfried et al., 2012; Sorg et al., 2014), which are feeding the desiccating Aral Sea, which now became itself a strong dust source (Xi and Sokolik, 2016). On the other hand, deposited dust and anthropogenic black carbon themselves can accelerate glacier melt by altering the glacier's surface albedo (Gabbi et al., 2015; Schmale et al., 2017).

The fieldwork of CADEX included column-integrating sun photometer measurements, vertically resolved lidar measurements, and meteorological observations in Dushanbe, Tajikistan. The lidar observations were carried out in Dushanbe at the Poligon measurement site of the Physical Technical Institute of the Academy of Sciences of Tajikistan. The measurement site lies in an urban environment on a hill in the eastern part of Dushanbe. The lidar is installed in front of a laboratory and office building (Fig. 2a). Its coordinates are $38°33'34''$ N, $68°51'22''$ E, and its altitude is $864$ m ASL. The measurement campaign lasted

from 17 March 2015 until 31 August 2016. During that period at 487 days lidar data has been acquired for a time period of at least 3 h. On 308 of these days the lidar ran even longer than 20 h.

In addition, the field experiment is accompanied by 3-D regional modeling with the regional dust model COSMO-MUSCAT (Wolke et al., 2012; Heinold et al., 2011) and the aerosol-climate model ECHAM-HAM (Zhang et al., 2012; Heinold et al., 2016). The modeling provides a perspective on the sources, transport, as well as the direct radiative effects of measured mineral

dust and associated atmospheric feedbacks. Furthermore, ground-based in situ aerosol measurements were conducted during CADEX to investigate the relationship between the settling of mineral dust along the margin of the Central Asian mountains and the dust transport at high tropospheric levels recorded by lidar. Collection of aerosol (March 2015 to April 2016) was carried out on quartz fiber filters (MK 360, MUNKTELL) using a high-volume filter sampler (DHA-80, DIGITEL) with an inlet for $PM_{10}$ (particulate matter with a maximum diameter of 10 μm). Chemical analysis of the collected aerosol was performed

with the techniques described in Fomba et al. (2014). The particle number size distribution was measured with a laser particle counter (GRIMM EDM 180) (Schettler et al., 2014). Dust dry deposition was collected (9 days in August 2016) by means of a flat plate type passive particle collector (Ott and Peters, 2008). Collected particles were subject to electron microscopy single particle analysis (Kandler et al., 2009, 2011), yielding information on particle size distribution, chemical and mineralogical composition as well as mixing state for coarse mode particles with diameter between 0.7 μm and 60 μm. The additional model

and in situ observational data, however, need further analysis and evaluation and will be used in a a follow-up study.

## 2.1 Lidar system

The lidar operated in Dushanbe is a multiwavelength polarization/Raman lidar Polly$^{XT}$ (POrtabLe Lidar sYstem Polly$^{XT}$, XT stands for extended version, Althausen et al. (2009)) and belongs to PollyNET, a network of permanent or temporary Polly systems (Baars et al., 2016). This specific Polly$^{XT}$ was already deployed in numerous field campaigns in the past (e.g. Baars

et al., 2012, 2016). For the CADEX campaign this Polly$^{XT}$ was upgraded with a second depolarization channel at 355 nm wavelength (Engelmann et al., 2016). All Polly$^{XT}$ systems contain a laser system that emits light at 355 nm, 532 nm, and 1064 nm wavelength by means of frequency doubling and tripling. The receiver of the used system has eight channels and measures the backscattered light at all three emitted wavelengths. The channels at 387 nm, 607 nm, and 407 nm wavelength allow to detect Raman scattering at night time. Another two channels detect cross-polarized light at 355 nm and 532 nm

wavelength. Three times a day, the system performs an automatic $\Delta 90°$-depolarization-calibration (Freudenthaler et al., 2009; Freudenthaler, 2016). Figure 2b shows a sketch of the optical layout of the used system. Signals are sampled with a vertical resolution of 7.5 m and are stored with a temporal resolution of 30 s. Full overlap is reached at about 1.5 km AGL (Baars et al., 2012). The resulting products of the Polly$^{XT}$ are summarized in Table 1. The corresponding uncertainties, discussed in detail in Freudenthaler et al. (2009), Baars et al. (2012, 2016) and Engelmann et al. (2016), result from uncertain input parameters and signal noise and are given as error bars in the results section. The polarization lidar photometer networking method (POLIPHON, (Ansmann et al., 2011b, 2012)) is applied to separate dust and non-dust optical and related microphysical properties. This separation is performed with respect to the particle backscatter coefficient (Tesche et al., 2009a; Groß et al., 2011; Mamouri and Ansmann, 2014) using the measured total particle linear depolarization ratio at 532 nm wavelength as well as typical particle linear depolarization ratio values (Tesche et al., 2009a; Müller et al., 2007) for dust (0.31) and non-dust (0.05). For the conversion of particle extinction coefficients into particle volume concentration, conversion factors for dust ($0.61 \cdot 10^{-6}$ m) and non-dust ($0.41 \cdot 10^{-6}$ m) from extended AERONET data analysis from Mamouri and Ansmann (2017) are used. Together with the densities of dust ($2.6\ \mathrm{g\,cm^{-3}}$) and non-dust ($1.6\ \mathrm{g\,cm^{-3}}$), this allows to calculate dust mass concentration profiles using characteristic lidar ratio values of dust (40 sr) and non-dust (80 sr). The uncertainty of such an estimation is given as 40% (Ansmann et al., 2011b). Bravo-Aranda et al. (2015) compared the POLIPHON method directly to in situ aircraft measurements and found less than 30% relative difference. The measured lidar data is uploaded daily and processed automatically (Baars et al., 2016) to be displayed as quicklooks on the PollyNET homepage (http://polly.rsd.tropos.de/).

## 2.2 Auxiliary data

Co-located with the lidar, a sun photometer as part of the AErosol RObotic NETwork (AERONET) (Holben et al., 1998) was operated. The Dushanbe AERONET station (http://aeronet.gsfc.nasa.gov/cgi-bin/type_one_station_opera_v2_new?site=Dushanbe) is operational since 2010 (Abdullaev et al., 2012). The CIMEL sun photometer measures at eight wavelengths (1020, 870, 675, 500, 490, 440, 380, and 340 nm) and retrieves the AOT and further column integrated particle optical and microphysical properties.

To calculate the Rayleigh contribution to the lidar signals in order to obtain particle optical properties, profiles of temperature and pressure are needed. Profiles of the GDAS (Global Data Assimilation System) with 1° spatial resolution from the National Weather Service's National Centers for Environmental Prediction (NCEP) at the coordinates 39° N, 69° E were used (GDAS, 2017), because no radiosonde launches were performed in Tajikistan since several years.

Publicly available trajectory models were used to assess the possible source regions and transport routes of the dust. The HYSPLIT model (Hybrid Single Particle Lagrangian Integrated Trajectory Model) (Stein et al., 2015; Rolph, 2016) as well as the FLEXPART model (FLEXible PARTicle dispersion model) version 9 (Stohl et al., 2005) were run to calculate backward trajectories for the presented example cases. The 0.5° GDAS meteorology for HYSPLIT and the 1° GFS (Global Forecast System) meteorology for FLEXPART is used. The HYSPLIT and FLEXPART backward trajectories were calculated with a starting time approximately corresponding to the time of observation of the aerosol layer by lidar. The arrival heights above the measurement site were set to the base, center, and top of the observed layer. The FLEXPART backward trajectories were

calculated for the arrival height above the measurement site in the height range of the aerosol layer measured with the lidar. The model output is the common logarithm of the accumulated residence time of air masses below 3 km during the model run time (Schwarz, 2015). This 3 km height is chosen because air is likely loaded with dust in the source regions, where planetary boundary layers of this magnitude occur. The spatial resolution of the model output is 1°.

## 3 Observations

### 3.1 Overview

An overview of the daily means of the AOT at 500 nm wavelength and the Ångström exponent from the 440–870 nm spectral range is shown in Fig. 3. The data were measured by sun photometer from March 2015 until August 2016 (cloudscreened AERONET data, Level 1.5). For the six 3-months seasons of the measurement campaign the means of the AOT, the Ångström
exponent and the fine mode fraction from the 500 nm measurements are shown in Tab. 2. A seasonal transition is obvious from the winter to the summer months. The AOT increases from spring to summer and decreases again in autumn. The Ångström exponent and the fine mode fraction behave reversely. There is a dusty 6-months season from April to September, and a less dusty season from October-March, but a strong dust event still occurred in October 2015. Within these seasons strong variations in the AOT and especially in the Ångström exponent occur from day to day. This indicates fast changes from dust dominated
to pollution dominated aerosol conditions.

Gkikas et al. (2009, 2013, 2016) and Georgoulias et al. (2016) classified dust events in the Mediterranean region based on the long-term mean of the AOT measured by satellite and ground stations. According to this classification, a strong dust event equals or exceeds the mean AOT plus 2 times the standard deviation, while an extreme dust event equals or exceeds the mean AOT plus 4 times the standard deviation. This classification can be pursued for the station in Dushanbe with AERONET data
(Level 2.0), which are available from July 2010 to November 2016 (1413 days). The mean of the daily mean AOT at 500 nm wavelength is 0.28 with a standard deviation of 0.23. As a comparison, Georgoulias et al. (2016) found a long-term mean AOT at 550 nm wavelength of $0.22 \pm 0.19$ for the Eastern Mediterranean region. So in Dushanbe, days with an AOT $\geqslant 0.74$ are therefore strong dust events, while days with an AOT $\geqslant 1.2$ at 500 nm wavelength are extreme dust events. Considering only at the measurement period from March 2015 to August 2016, the mean AOT at 500 nm wavelength was 0.28 (with a standard
deviation of 0.16) and therefore equal to the long-term mean. According to this classification, 6 episodes of strong dust and 6 episodes of extreme dust occurred during the measurement period. In the period July 2010 to November 2016 (6.25 years) strong dust events occurred 19 times (3 per year), extreme dust events 10 times (1.6 per year) and more than extreme dust events with an AOT above 2 at 500 nm wavelength occurred 4 times (0.6 per year).

### 3.2 Illustrative measurement examples

To illustrate the frequently observed variety and complexity of dust layers above Tajikistan, examples of dust layers of various origin occurring at all heights from the surface to cirrus level are presented in Fig. 4. Figure 4a shows the temporal development

of the range-corrected signal of the 1064 nm wavelength channel from 20–24 April 2015. During these four days multiple dust layers arrived above Tajikistan. On 20 April 2015 there were aerosol layers up to 4 km height. At the same time, descending dust layers crossed the lidar station at heights of about 10 km. On 22 April 2015 a second dust layer arrived at 10 km height, which again descended. According to FLEXPART trajectories, the Arabian Peninsula, Iran, and also parts of Uzbekistan and

Turkmenistan were source regions for the detected dust (Fig. 4d). A high altitude aerosol layer was measured on 13 May 2015, reaching roughly from 6.5 km to 10 km altitude (Fig. 4b). FLEXPART backtrajectory analysis shows that this high layer was long-range transported towards the measurement site from the Mediterranean/North African region (Fig. 4e). The third scenario is a near-ground dust layer (Fig. 4c, described in detail in Sec. 3.3.2), which was measured on 8 and 9 August 2015 and had Central Asian sources (Fig. 4f).

## 3.3 Case studies

In the following sections we discuss four strongly contrasting cases in more detail: (1) A lofted layer of Middle Eastern dust that occurred on 13 April 2015 (Case 1, Fig. 5, section 3.3.1), (2) an extreme dust event with Central Asian dust, which was recorded on 8 and 9 August 2015 (Case 2, Fig. 6, section 3.3.2), (3) the most extreme dust event during the CADEX campaign with dust from Central Asian sources, which was observed on 14 July 2016 (Case 3, Fig. 7, section 3.3.3), and (4) a contrasting

case with a pollution layer of local origin, which was measured on 13 May 2015 (Case 4, Fig. 8, section 3.3.4).

### 3.3.1 Case 1: 13 April 2015, lofted dust layer

Figure 5a shows the temporal development of the range-corrected signal of the 1064 nm wavelength channel on 12–13 April 2015. The dust layer arrived over Dushanbe on 12 April 2015 at an altitude between 5 km and 6 km. The slowly descending dust layer (red) contained dense clouds (grey). On 13 April 2015 the dust layer extended at about 2.5–5 km height. Its base was

very sharp while on top of the layer a thinner aerosol layer was measured up to 7 km. At the end of that day, a down-mixing of dust into the lowermost 2 km occurred. The HYSPLIT backward trajectories arriving on 13 April 2015 show that the air parcels are coming from southwestern direction (Fig. 5d). The upper and the lower trajectories come from the Arabian Peninsula and travel over central Iran and along the Afghanistan-Turkmenistan border. The upper trajectory is always more than 3 km above ground, while the lower trajectory reached below 2 km once above the Arabian Peninsula. The center trajectory is coming

from more southern direction, traveling over southern Iran, where it reaches lowest down to below 1 km above the Iranian Dashte-Lut desert.

Similar to HYSPLIT, the FLEXPART backward trajectories indicate Iran including its southern parts and large parts of the Arabian Peninsula as source regions (Fig. 5c). In contrast to HYSPLIT, the FLEXPART backward trajectories resided partly above Afghanistan.

Vertical profiles of the optical properties of this dust layer are shown in Fig. 5b. The particle linear depolarization ratio at 532 nm wavelength is on average 0.31 and 0.34 at its maximum. This shows that this lofted dust layer consists of almost unpolluted dust. Depolarization ratios of pure Middle Eastern dust close to its source regions have been found as 0.3–0.35 at 532 nm wavelength (Mamouri et al., 2013, 2016). The mean lidar ratio is $35.7 \pm 1.7$ sr at 532 nm wavelength, what is

comparable to measurements for Middle Eastern dust (Mamouri et al., 2013; Nisantzi et al., 2015). The lofted dust layer has a maximum dust mass concentration of $196\,\mathrm{\mu g\,m^{-3}}$ at the bottom of the layer at 2.7 km. In the middle of the layer at 3.5 km the dust mass concentration reaches $171\,\mathrm{\mu g\,m^{-3}}$. The integrated dust mass up to 5 km altitude is $0.51\,\mathrm{g\,m^{-2}}$. The non-dust components in this layer are negligible, except in the upper parts of the layer, where the particle linear depolarization ratio

partly drops below 0.31 at 532 nm wavelength.

### 3.3.2    Case 2: 8/9 August 2015, extreme dust event

An extreme dust event was measured on 8 and 9 August 2015. Figure 4c shows the temporal development of the range-corrected signal of the 1064 nm wavelength channel on 8–9 August 2015. The dust got lifted up with the evolution of the convective boundary layer. This lead to a very strong signal in the lowest 1.5 km with dust extending up to about 3 km height.

The FLEXPART trajectories arriving on 8 August 2015 (Fig. 4f) exhibit a large area northwest of Tajikistan with high accumulated residence times, even west of the Caspian Sea.

Vertical profiles of the optical properties of this dust layer are shown in Fig. 6. The particle extinction coefficients at 355 nm and 532 nm wavelength are about $550\,\mathrm{Mm^{-1}}$ at 1 km height. Apparently, the layer is divided in two parts. There is a maximum in the first 1 km above ground and then again at about 2.5 km height. But looking at the intensive properties, especially the

particle linear depolarization ratio and the extinction-related Ångström exponent, continuous features are recognizable. The twofold layer probably contains the same aerosol and originates from the same source region since it has almost identical optical properties. The apparent layer boundary at 1.6 km is probably formed by the diurnal cycle of the boundary layer. The development of the convective boundary layer after about 06:00 UTC on 8 August 2015 is visible in Fig. 4c. The mean particle linear depolarization ratios are higher than in the lofted dust layer before (Case 1), on average 0.35 at 532 nm wavelength. The

maximum dust mass concentration of that extreme near-ground dust layer is $845\,\mathrm{\mu g\,m^{-3}}$ at 0.65 km height, in the middle of the layer the minimum is $475\,\mathrm{\mu g\,m^{-3}}$ at 1.7 km. In the upper part at 2.4 km the dust mass concentration is again $663\,\mathrm{\mu g\,m^{-3}}$. The integrated dust mass up to 4 km altitude is $3.2\,\mathrm{g\,m^{-2}}$.

### 3.3.3    Case 3: 14 July 2016, most extreme dust event

The most extreme dust event during the CADEX campaign occurred on 14 July 2016. The dust persisted during the next four

days (Fig. 3). The FLEXPART trajectories arriving on 14 July 2016 (Fig. 7b) show large accumulated residence times as far away as western Iran, but the highest values are in Uzbekistan. The accumulated residence times above Kazakhstan are high, but they are also reaching eastwards towards the Lake Balkhash, differently than on 8 August 2015 (Fig. 4f). The HYSPLIT backward trajectories arriving on 14 July 2016 are shown in Fig. 7c. The higher trajectory arrived from the Aral Sea and the lower one from central Kazakhstan through Turkmenistan and Afghanistan. The vertical profiles of the measured optical

properties of this dust layer are presented in Fig. 7a. The particle extinction coefficients at 355 nm and 532 nm wavelength are about $1.7\,\mathrm{km^{-1}}$ at 1 km height. The resulting AOT is 3.89 at 532 nm wavelength, what is much higher than the long-term mean plus 10 times the standard deviation (see section 3.1). The 14 July 2016 has the highest daily mean AOT since the beginning of the record in Dushanbe with the AERONET sun photometer in 2010. So this dust event could well be called record-breaking

like Mamouri et al. (2016) called an extreme dust event in September 2015 in the Mediterranean. The mean particle linear depolarization ratio at 355 nm wavelength is 0.29 and thus higher than in Case 2, while at 532 nm wavelength it is equally 0.35. The calculated dust mass concentrations for that most extreme dust is largest at 1 km altitude with 2.8 $\mathrm{mg\,m^{-3}}$. The integrated dust mass up to 2.7 km altitude is 6.5 $\mathrm{g\,m^{-2}}$.

### 3.3.4 Case 4: 13 May 2015, contrasting case with local pollution

Figure 4b shows the temporal development of the range-corrected signal of the 1064 nm wavelength channel on 13–14 May 2015. There were several aerosol layers distributed up to 10 km height. The HYSPLIT backward trajectories arriving on 13 May 2015 are shown in Fig. 8b. The upper trajectory is coming from northwest of Tajikistan and the lower trajectories reach down to the ground the same day inside Tajikistan, indicating the contribution of local pollution to the lower altitude aerosol layers.

The vertical profiles of the measured optical properties are presented in Fig. 8a. The particle extinction coefficients in the layer at about 2–4 km are about 40 $\mathrm{Mm^{-1}}$ at 355 nm and 20 $\mathrm{Mm^{-1}}$ at 532 nm wavelength. This aerosol layer is barely depolarizing and the mean extinction-related Ångström exponent is $2.07 \pm 0.72$. The dust mass concentration reaches 6.9 $\mathrm{\mu g\,m^{-3}}$ at 2.5 km height, the non-dust mass concentration is at the same time dominating with 30.7 $\mathrm{\mu g\,m^{-3}}$. The integrated dust mass up to 10.4 km altitude is 0.05 $\mathrm{g\,m^{-2}}$ and the integrated non-dust mass is 0.13 $\mathrm{g\,m^{-2}}$. Finally, an overview over the individual cases is given in Tab. 3, where the intensive optical properties and AOTs of the presented example cases are summarized.

## 4 Discussion

Four exemplary measurement cases are described in this article. Two of them are from spring and one is from summer 2015 and one is from summer 2016. In early spring, the AOT was predominantly low ($\leq 0.3$) (Fig. 3) but a lot of dust layers occurred (e.g. 13 April 2015 (Case 1, Fig. 5), 20–24 April 2015 (Fig. 4a). In summer, the AOT in general was higher ($\geq 0.35$) and some dust events overtopped that clearly (e.g. 8-9 August 2015 (Case 2, Fig. 6), 14 July 2016 (Case 3, Fig. 7)). However, a statistical analysis of the lidar profiles of the whole measurement period has yet to follow.

### 4.1 Dust AOT and mass concentration

The lofted dust layer in Case 1 contributed significantly to the total AOT of about 0.4 at 532 nm wavelength. Case 2 had an AOT of 1.5 at 532 nm wavelength. The most extreme dust case during the measurement campaign was Case 3 with an extraordinary AOT of above 3.5 at 532 nm wavelength measured by lidar, respectively above 4 by sun photometer. The AOT of Case 2 is comparable to a dust event during the Soviet-American campaign in 1989 (AOT of 1.3 at 550 nm wavelength) (Gomes and Gillette, 1993). The most extreme dust event (Case 3) is comparable to the second dust event during the Soviet-American campaign (AOT of 3.3 at 550 nm wavelength) (Pachenko et al., 1993). Pachenko et al. (1993) estimated the maximum AOT of that dust event a day before was greater than 10, based on horizontal visibility, which was reported to be 50–200 m (Smirnov et al., 1994). As a comparison, Mamouri et al. (2016) reported a visibility of 500–600 m for an extraordinary dust event in the

Mediterranean area, inferring an AOT of 4.8 to 9 depending on the layer height. This again lead to an estimate of the column dust load of 8–15 $\mathrm{g\,m^{-2}}$. These values are higher than for Case 3, where the AOT was 3.89 at 532 nm wavelength, and the integrated dust mass was 6.7 $\mathrm{g\,m^{-2}}$. The visibility calculated from the measured particle extinction coefficient of 1.7 $\mathrm{km^{-1}}$ (measured at 0.87 km altitude) was 2.3 km. The AOT and the dust mass concentrations for this Case 3 were calculated based on the lidar data only, but the sun photometer measured even a higher AOT of 4.45 at 500 nm wavelength.

## 4.2 Dust layer height

The multiple high altitude dust layers preceding the lofted dust layer on 24 April 2015 (Fig. 4a) as well as the high aerosol layer on 13 May 2015 (Fig. 4b) reached heights of about 10 km AGL. This is higher than the highest point of the Pamir mountains (7.6 km ASL), which means that these layers can cross the Pamir or the Tienshan mountains (7.4 km ASL) and can be transported further eastwards. A similar observation was made in spring 2003 when layers of elevated depolarization at 9–11 km height have been measured by lidar in Aksu (northwestern Taklamakan) (Mikami et al., 2006). Within a week time, lidar stations in Japan measured dust at altitudes between 2 km and 6 km, without having major dust outbreaks in the Taklamakan and Gobi deserts. Model simulations indicated that the dust was transported to Japan via north of the Tienshan mountains (Tanaka et al., 2005). For that case, Tanaka et al. (2005) estimated 50% of the dust particles arriving in Japan came from the Sahara, 30% from the Middle East, and only 10% from China.

## 4.3 Lidar ratio and dust source region

The observed lidar ratios in the presented dust cases (Cases 1,2,3) range from 40.3 sr to 46.9 sr at 355 nm and from 35.7 sr to 42.9 sr at 532 nm wavelength. The lofted dust layer (Case 1) has lidar ratios of 42.2 sr at 355 nm and 35.7 sr at 532 nm wavelength. The near-ground dust layers (Cases 2,3) have lidar ratios of 46.9 sr and 40.3 sr at 355 nm and 42.9 sr and 38.7 sr at 532 nm wavelength. The only weakly depolarizing pollution aerosol in Case 4 has lidar ratios of 55.8 sr at 355 nm and 32.8 sr at 532 nm wavelength, but with high variability.

Saharan dust is found to have lidar ratios of 50–60 sr at 532 nm wavelength (e.g. Groß et al., 2015; Tesche et al., 2009b). Schuster et al. (2012) used AERONET data sets to retrieve pure dust lidar ratios and found lower values for West-Asian dust than for North African dust. Similar results were found from Raman lidar measurements in Cyprus with lidar ratios of 35–45 sr for Middle Eastern dust (Mamouri et al., 2013; Nisantzi et al., 2015). In east Asia, lidar ratios of Asian dust of for example $47 \pm 18$ sr (Sakai et al., 2003) and 42–73 sr (Liu et al., 2002) at 532 nm wavelength were reported. In China, lidar ratios of $40 \pm 5$ sr (Tesche et al., 2007) and $35 \pm 5$ sr (Müller et al., 2007) at 532 nm wavelength for dust from the Gobi desert were measured. The situation in Central Asia is even more unclear, as almost no measurements exist. Although direct measurements of dust lidar ratios inside of the Taklakaman are not yet available, Jin et al. (2010) used constrains to retrieve a dust lidar ratio of $42 \pm 3$ sr at 532 nm wavelength in Aksu (northwestern Taklamakan). Chen et al. (2013b) measured very low lidar ratios of 8–29 sr at 532 nm wavelength in weakly depolarizing dust layers in Kyrgyzstan. For dust from the Caspian and Aral sea region, a lidar ratio of $43 \pm 3$ sr at 532 nm was measured (Dieudonné et al., 2015).

The lidar ratios of the example dust cases (Cases 1,2,3) agree well with these values. There is no clear difference between the

lofted dust layer (Case 1), which is long-range transported Middle Eastern dust and the near-ground fresh dust layers (Case 2,3) from Central Asian sources. The lofted dust layer in Case 1 has a very low lidar ratio of $35.7 \pm 1.7$ at 532 nm wavelength, indeed comparable to the ones ($33.7 \pm 6.7$ to $39.1 \pm 5.1$ sr at 532 nm) found by Mamouri et al. (2013) for dust from the Middle East. On the other hand, also the most extreme dust event (Case 3) has a lidar ratio below 40 sr at 532 nm wavelength.

Then again, slightly larger lidar ratios above 40 sr were measured in the extreme near-ground dust layer (Case 2). The lidar ratios of the pollution layer (Case 4) differ significantly between the two wavelengths.

## 4.4 Dust particle linear depolarization ratio

The observed depolarization ratios in the presented dust cases (Cases 1,2,3) range from 0.18 to 0.29 at 355 nm and from 0.31 to 0.35 at 532 nm wavelength. The lofted dust layer (Case 1) has a depolarization ratio of 0.18 at 355 nm and 0.31 at 532 nm

wavelength. The two near-ground dust cases (Cases 2,3) have depolarization ratios of 0.23 and 0.29 at 355 nm and both 0.35 at 532 nm wavelength. In contrast to the dust cases, the local pollution aerosol (Case 4) has a low depolarization ratio of 0.03 at 355 nm and 0.08 at 532 nm wavelength.

Dieudonné et al. (2015) measured a particle linear depolarization ratio of $0.23 \pm 0.02$ at 532 nm wavelength for lofted dust from the Caspian and Aral sea region. From the Kyrgyzstan station depolarization ratio values of 0.1–0.15 for lofted dust layers

(Chen et al., 2013b) and around 0.2 at 532 nm wavelength for near-ground dust layers were reported (Chen and Sverdlik, 2007). But it has to be considered that this station is already located at 1.9 km ASL. In Aksu (northwestern Taklamakan), Kai et al. (2008) measured depolarization ratios at 532 nm wavelength of $0.09^*$–$0.11^*$ in a lofted dust layer and $0.18^*$–$0.33^*$ in a near-ground dust layer. Iwasaka et al. (2003) measured a depolarization ratio of $0.27^*$ at 532 nm wavelength in a lofted dust layer in Dunhuang (northern Taklamakan). Note that the values denoted with $^*$ are published as aerosol depolarization potentials and

are converted to particle linear depolarization ratios (see Burton et al., 2014; Cairo et al., 1999; Gimmestad, 2008). The measured depolarization ratios for the presented dust cases (Cases 1,2,3) are mostly higher than these literature values for dust measured in or close to Central Asia. This suggests that those studies might have described also observations of polluted/mixed dust. The range of the presented particle depolarization ratios (0.31–0.35) at 532 nm wavelength is comparable with the values of fresh Saharan dust (0.27–0.35) (Freudenthaler et al., 2009) or Middle Eastern dust (0.25–0.32) (Mamouri

et al., 2016). The spectral difference between the wavelengths is considerable. The range of the presented particle depolarization ratios at 355 nm wavelength (0.18–0.29) is large, with the exceptionally high value during the most extreme dust event (Case 3). In Saharan dust, particle depolarization ratios at 355 nm wavelength in ranges of 0.22–0.31 were measured at the source region (Freudenthaler et al., 2009) and 0.21–0.27 after long-range transport (Burton et al., 2015; Groß et al., 2015; Haarig et al., 2016, 2017). The near-ground dust cases (Cases 2,3) had higher particle depolarization ratios at both wavelengths than the lofted dust

layer of Case 1. Nevertheless, the lofted dust layer in Case 1 had a depolarization ratio of 0.31 at 532 nm wavelength, which indicates only minor changes of Middle Eastern dust depolarization characteristics during its long-range transport towards Central Asia. On the other hand, the lofted dust layer in Case 1 had a lower depolarization ratio at 355 nm wavelength than long-range transported Saharan dust.

## 5 Conclusions

For the first time, multiwavelength polarization/Raman lidar observations have been carried out in Tajikistan in the framework of the CADEX campaign. The continuous 18-months measurement provide a unique data set of vertically resolved aerosol optical properties in Central Asia. Additional ground-based in situ measurements and modeling studies accompanied the CADEX campaign (not presented here). During the campaign, dust layers were observed frequently from the surface to tropopause heights, having source regions in Africa, Middle East and Central Asia. Four case studies which are typical for the conditions at the location have been presented, such as lofted dust layers and extreme dust events. As a contrast to the dust cases, an example of a pollution aerosol layer of local origin illustrates the non-negligible anthropogenic influence on the aerosol in Tajikistan. The observed particle linear depolarization ratios for the dust cases range from 0.18–0.29 at 355 nm and 0.31–0.35 at 532 nm wavelength. The presented examples of near-ground layers of Central Asian dust had high depolarization ratios in both wavelengths indicating pure dust conditions. The presented lofted dust layer of long-range transported Middle Eastern dust had slightly lower depolarization ratios, especially at 355 nm wavelength. Nevertheless, the values still indicate almost unpolluted dust conditions after long-range transport to Central Asia. The observed lidar ratios in the presented dust cases range from 40.3 sr to 46.9 sr at 355 nm and from 35.7 sr to 42.9 sr at 532 nm wavelength. These lidar ratio values are lower than typical lidar ratios of Saharan dust (50–60 sr) and comparable to lidar ratios of dust from Middle East/West-Asia (35–45 sr). Further analyses of the data set, to be published in a follow-up publication, include a statistical analysis of the whole measurement period regarding the optical properties, dust mass concentrations, dust layer heights, seasons, and source regions.

Nevertheless, more measurements in Central Asia are needed to analyze long-term trends, especially with respect to climate change. Such measurements offer also new satellite comparison possibilities. Moreover, the assimilation of these data into dust models (regional and global transport, aerosol optical properties, radiative transfer) will help to reduce uncertainties. Therefore, a permanent lidar station in Tajikistan will be established starting most likely from 2019.

## 6 Data availability

HYSPLIT backward trajectories are calculated via the available online tools (http://ready.arl.noaa.gov/HYSPLIT.php). AERONET sun photometer data are available from the AERONET web page (http://aeronet.gsfc.nasa.gov/cgi-bin/type_one_station_opera_v2_new?site=Dushanbe). Data of the FLEXPART backward trajectory calculations and the CADEX Polly$^{XT}$ lidar data are available at Leibniz Institute for Tropospheric Research, Leipzig.

*Acknowledgements.* The CADEX project was funded by the German Federal Ministry of Education and Research (BMBF) in the context of "Partnerships for sustainable problem solving in emerging and developing countries" under the grant number 01DK14014. This project has also received funding from the European Union's Horizon 2020 research and innovation programme under grant agreement No 654109. We like to thank the Academy of Sciences of Tajikistan for the helpful support of any kind during all stages of the CADEX campaign. We like to thank Lars Klüser from German Aerospace Center, Emmanouil Proestakis from National Observatory of Athens, Alexandra Chud-

novsky from University of Tel Aviv for the opportunity to compare the ground-based measurements to satellite measurements. K. Kandler acknowledges support from the Deutsche Forschungsgemeinschaft (grant KA2280/2).

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

**Table 1.** Products of the Polly$^{XT}$ lidar and their relative uncertainties. $\beta$: particle backscatter coefficient, $\alpha$: particle extinction coefficient, $S$: lidar ratio, $\mathring{\alpha}$: extinction-related Ångström exponent, $\delta$: particle linear depolarization ratio, WV: water vapor, and dust and non-dust mass concentrations.

| Product | Wavelengths [nm] | Uncertainty [%] |
|---|---|---|
| $\beta$ | 355, 532, 1064 | 5-10 |
| $\alpha$ | 355 (from 387), 532 (from 607) | 10-20 |
| $S$ | 355, 532 | 11-22 |
| $\mathring{\alpha}$ | from $\alpha$ 355/532 pair | 14-28 |
| $\delta$ | 355, 532 | 7-11 |
| WV | from 407 | 20 |
| Mass conc. | from $\delta$, $\beta$, 355, 532 | 30-50 |

**Table 2.** Seasonal means of the sun photometer measurements (AERONET Level 1.5, cloudscreened). AOT at 500 nm wavelength, Ångström exponent (AE) from the 440-870 nm spectral range, and the fine mode fraction (FMF) from 500 nm wavelength.

| Season | AOT | AE | FMF |
|---|---|---|---|
| Mar–May 2015 | 0.16 | 0.63 | 0.39 |
| Jun–Aug 2015 | 0.40 | 0.54 | 0.33 |
| Sep–Nov 2015 | 0.23 | 1.03 | 0.51 |
| Dec–Feb 2015/16 | 0.15 | 1.28 | 0.71 |
| Mar–May 2016 | 0.22 | 0.82 | 0.48 |
| Jun–Aug 2016 | 0.41 | 0.52 | 0.34 |

**Table 3.** Overview over the selected example cases. $S_{355}, S_{532}$: Lidar ratio at 355 nm and 532 nm wavelength. $\delta^p_{355}, \delta^p_{532}$: Particle linear depolarization ratio at 355 nm and 532 nm wavelength. $\tau_{lidar}$: Aerosol optical thickness measured with lidar at 532 nm wavelength. $\tau_{SPM}$: Aerosol optical thickness measurement of the co-located sun photometer at 500 nm wavelength (time of measurements in the footnote). $\mathring{\alpha}_{355/532}$: Extinction-related Ångström exponent based on the particle extinction coefficients at 355 nm and 532 nm wavelength. $\mathring{\beta}_{355/532}, \mathring{\beta}_{532/1064}$: Backscatter-related Ångström exponents based on the particle backscatter coefficients at 355 nm, 532 nm, and 1064 nm wavelength. The height range to average the optical properties of the dust layers is placed within the core of each layer. The provided uncertainties are the standard deviations of the averaged values.

| Case | Case 1 Lofted dust | Case 2 Extreme dust | Case 3 Most extreme dust | Case 4 Pollution |
|---|---|---|---|---|
| Source region | Middle East | Central Asia | Central Asia | Local |
| Date | 13 April 2015 | 8 August 2015 | 14 July 2016 | 13 May 2015 |
| Time [UTC] | 15:10–16:09 | 22:20–23:57 | 16:00–21:59 | 18:10–22:00 |
| Layer height [km AGL] | 2.3–4.9 | 0–3.1 | 0–2.6 | 0–4.1 |
| $S_{355}$ [sr] | $42.2 \pm 3.0$ | $46.9 \pm 2.1$ | $40.3 \pm 0.6$ | $55.8 \pm 7.1$ |
| $S_{532}$ [sr] | $35.7 \pm 1.7$ | $42.9 \pm 3.2$ | $38.7 \pm 1.0$ | $32.8 \pm 6.4$ |
| $\delta^p_{355}$ | $0.18 \pm 0.02$ | $0.23 \pm 0.01$ | $0.29 \pm 0.01$ | $0.03 \pm 0.01$ |
| $\delta^p_{532}$ | $0.31 \pm 0.01$ | $0.35 \pm 0.01$ | $0.35 \pm 0.01$ | $0.08 \pm 0.01$ |
| $\tau_{lidar}$ | 0.40 (0-5.4 km) | 1.50 (0-4.1 km) | 3.89 (0-3 km) | 0.13 (0-4.5 km) |
| $\tau_{SPM}$ | $0.41^{(1)}$ | $1.19^{(2)}, 1.69^{(3)}$ | $4.45^{(4)}, 2.89^{(5)}$ | $0.18^{(6)}$ |
| $\mathring{\alpha}_{355/532}$ | $0.41 \pm 0.24$ | $0.12 \pm 0.16$ | $-0.08 \pm 0.06$ | $2.07 \pm 0.72$ |
| $\mathring{\beta}_{355/532}$ | $0.00 \pm 0.22$ | $-0.11 \pm 0.09$ | $-0.20 \pm 0.13$ | $0.72 \pm 0.16$ |
| $\mathring{\beta}_{532/1064}$ | $0.12 \pm 0.04$ | $0.32 \pm 0.07$ | $0.29 \pm 0.03$ | $0.71 \pm 0.05$ |

Times of sun photometer measurements [UTC]: [1] 13 April 2015, 12:54:46, [2] 8 August 2015, 13:04:24, [3] 9 August 2015, 02:08:05, [4] 14 July 2016, 09:16:02, [5] 15 July 2016, 03:29:34, [6] 13 May 2015, 13:21:50

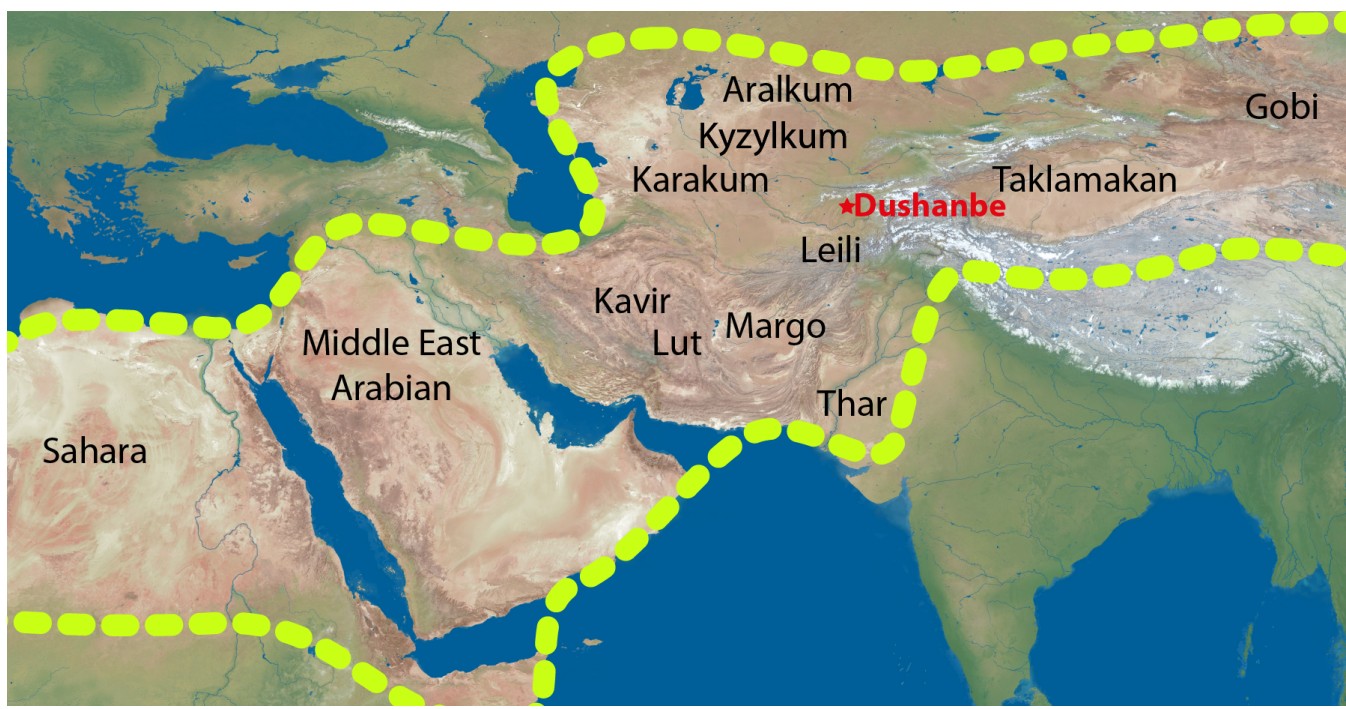

**Figure 1.** Geographical map with desert regions (black text), dust belt (bright green dotted line), measurement site in Dushanbe, Tajikistan (red text) highlighted (http://naturalearth.springercarto.com, adapted).

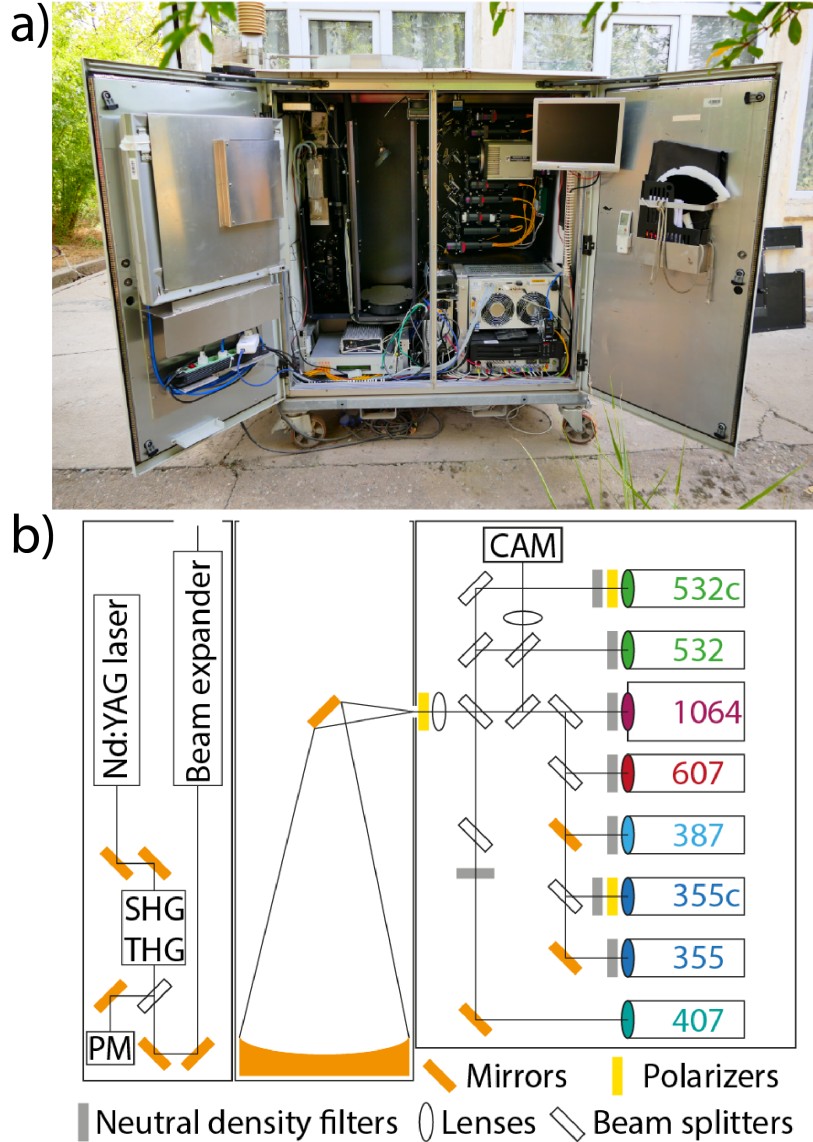

**Figure 2.** a) Polly$^{XT}$ with closed roof, open doors, and removed protective plates at the Poligon field site in Dushanbe. The cabinet size is approximately 1.9×1.7×0.9 m. b) Schematic sketch of the optical layout of the Polly$^{XT}$. Left (transmitter part): After the second harmonic generation (SHG) and the third harmonic generation (THG), parts of the laser beam are deflected to a power meter (PM) which measures the UV component to monitor the conversion efficiency. Right (receiver part): Backscattered light is collected with a Newtonian telescope and then passed towards the receiver unit. The numbers indicate the wavelength in nm of the detection channels and c denotes the cross-polarized channels. A camera (CAM) is synchronized to the laser trigger and sees the beam at 532 nm wavelength to monitor the overlap. The polarizer mounted in front of the pinhole is a device for the absolute calibration of the depolarization measurements (details see Engelmann et al., 2016).

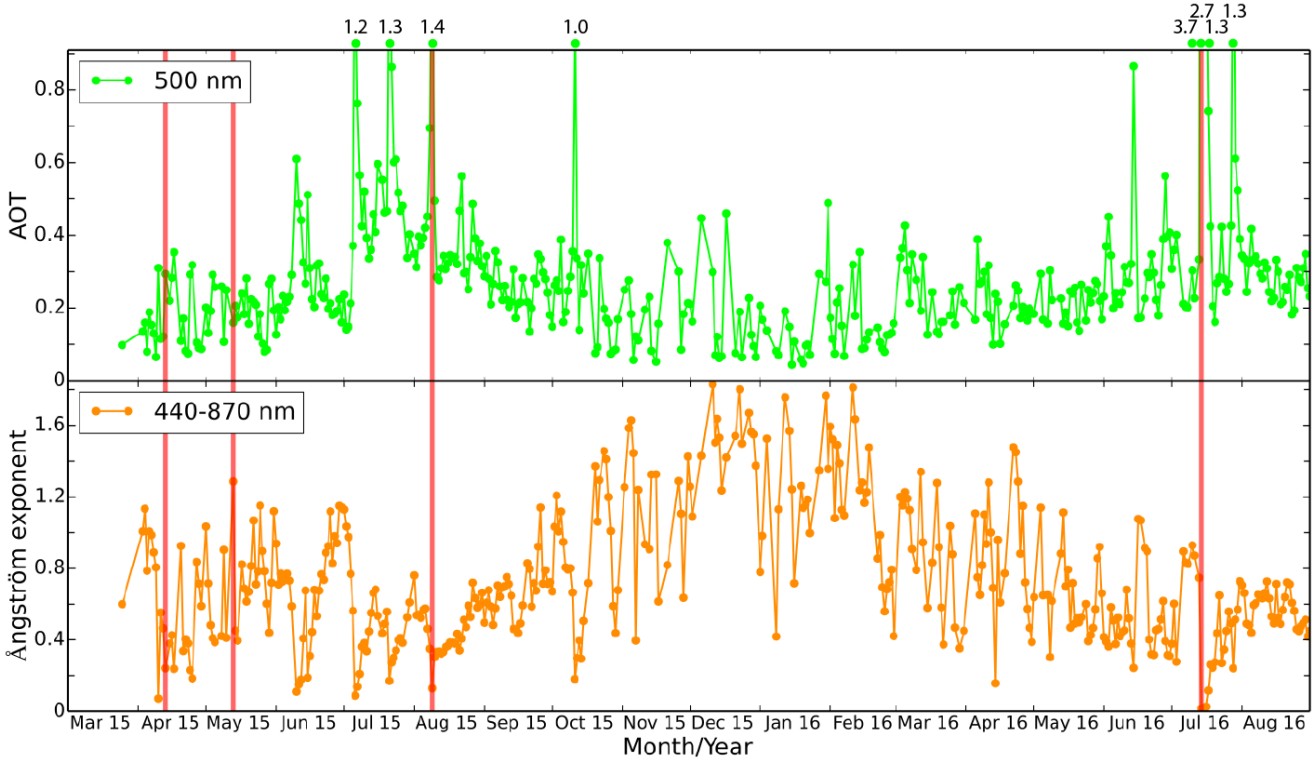

**Figure 3.** Daily means of the AOT (top) and Ångström exponent (bottom) measured in Dushanbe from March 2015 until August 2016 by sun photometer (AERONET Level 1.5, clouscreened). The red lines indicate the days of the presented example cases.

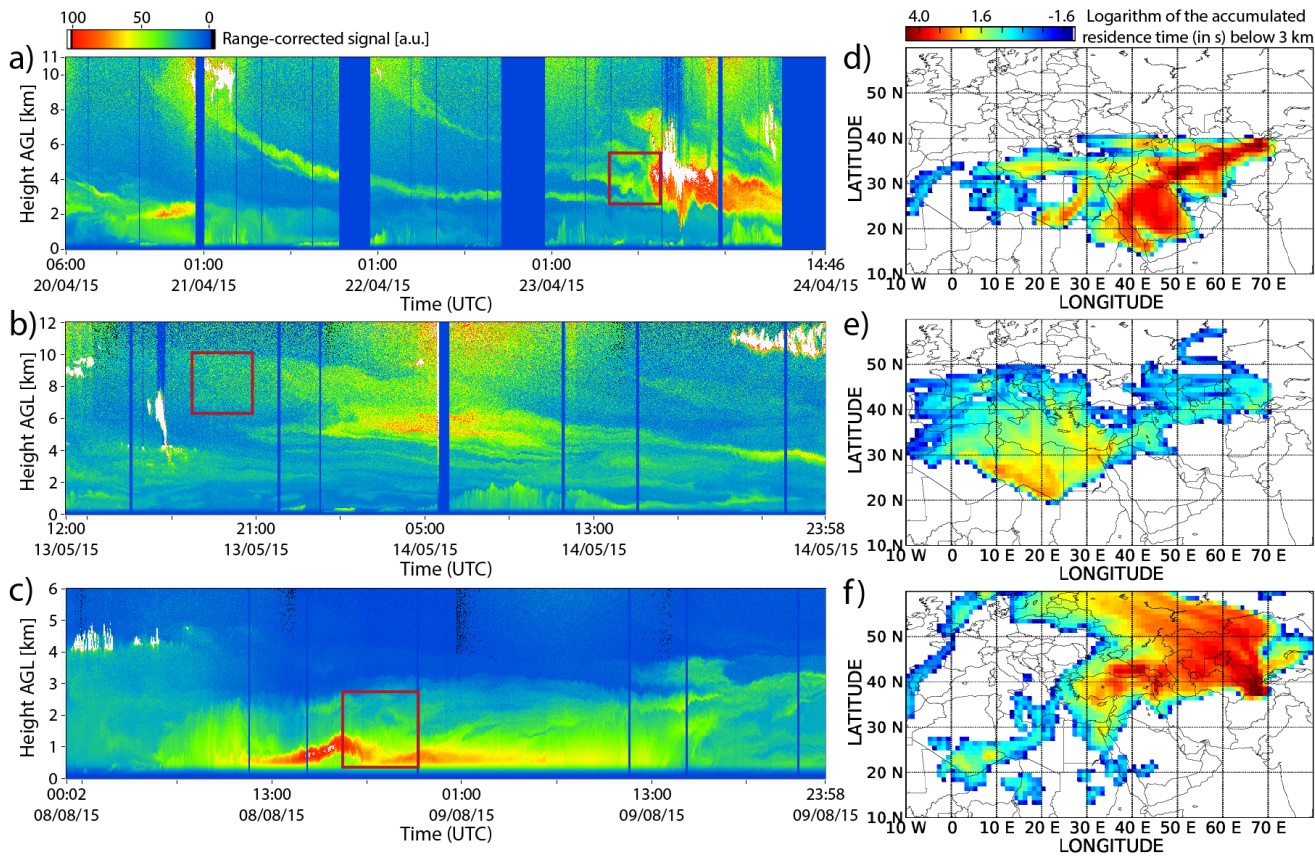

**Figure 4.** Left panels: Illustrative measurement examples of aerosol layers observed with Polly$^{XT}$ at Dushanbe, partly up to cirrus level (10 km height). Temporal development of the range-corrected signal (temporal resolution 30 s, vertical resolution 7.5 m) of the 1064 nm wavelength channel on a) 20 April 2015, 06:00 UTC – 24 April 2015, 14:46 UTC. b) 13 May 2015, 12:00 UTC – 14 May 2015, 23:58 UTC. c) 8 August 2015, 00:02 UTC – 9 August 2015, 23:58 UTC. Blue rectangles denote periods where no measurements were performed. Right panels: Source regions of the observed aerosol identified based on FLEXPART model runs. The first lofted layer (a) has Middle Eastern sources (d), the second lofted layer (b) has North African sources (e) and the third low laying dust (c) has Central Asian sources (f). The dark red squares in the left panels indicate the arrival height and time of the calculated backward trajectories in context of the lidar measurements. The right panels show maps of the logarithm of accumulated residence time below 3 km (in s) of 144 h FLEXPART backward trajectories arriving above Dushanbe on d) 23 April 2015, 20:30–21:30 UTC between 2.7 and 5 km, e) 13 May 2015, 17:30–18:30 UTC between 6.6 and 9.9 km height, f) 8 August 2015, 21:30–22:30 UTC between 0.5 and 3.1 km height.

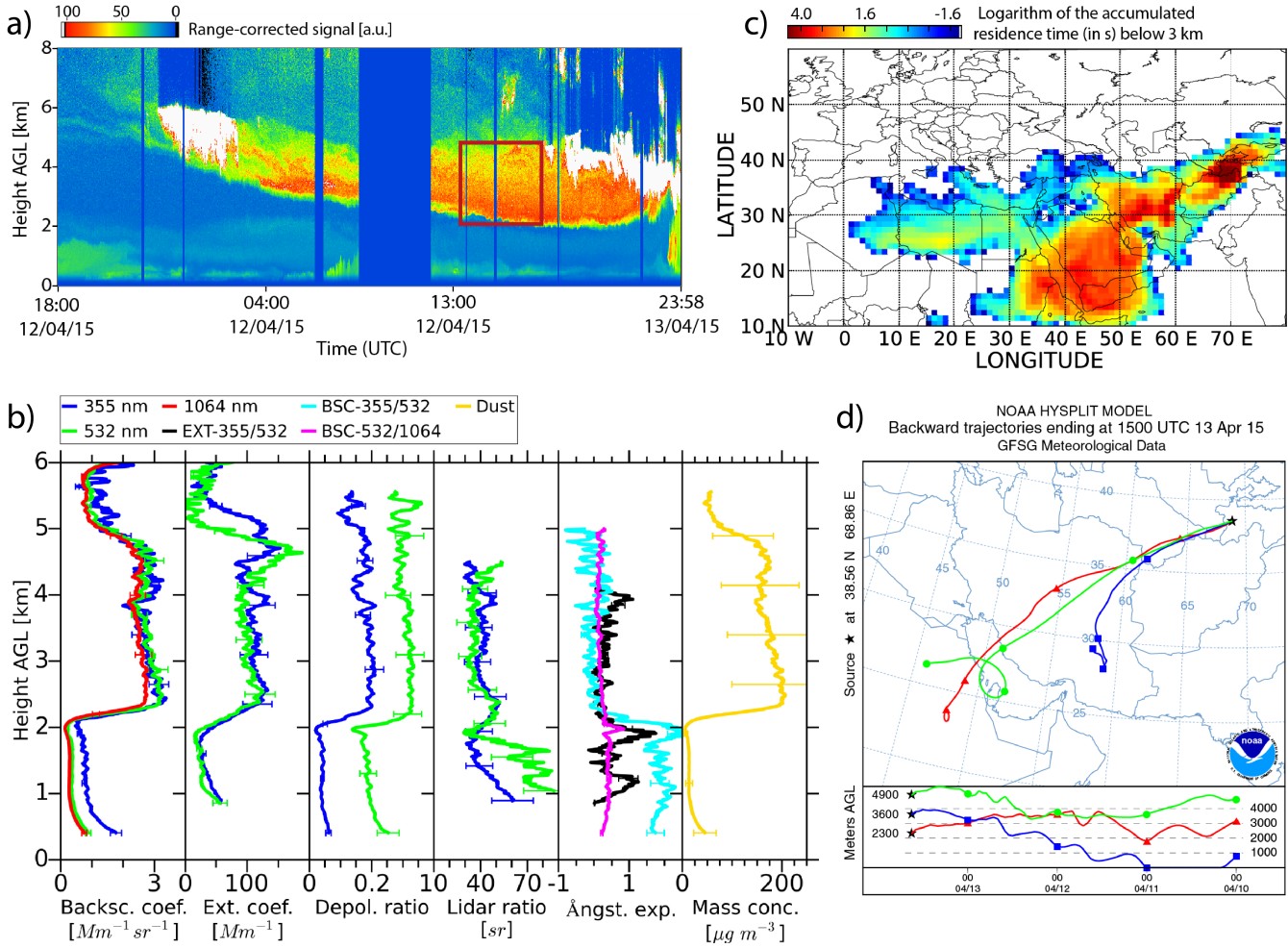

**Figure 5.** Case 1. A lofted layer of Middle Eastern dust is shown. a) Same as Fig. 4 (left panels), but on 12 April 2015, 18:00 UTC – 13 April 2015, 23:58 UTC. b) The averaged lidar profiles were measured on 13 April 2015, 15:10–16:08 UTC. Lidar signals are smoothed before calculation of the extinction-related (743 m) and backscatter-related (68 m) optical properties. Particle backscatter coefficient at 355 nm, 532 nm and 1064 nm wavelength. Particle extinction coefficient, particle linear depolarization ratio, and lidar ratio at 355 nm and 532 nm wavelength. Extinction-related Ångström exponent from 355 nm and 532 nm wavelengths (black), backscatter-related Ångström exponent from 355 nm and 532 nm wavelengths (aqua), and backscatter-related Ångström exponent from 532 nm and 1064 nm wavelengths (magenta). Dust mass concentration. c) 144 h FLEXPART backward trajectories same as in Fig. 4 (right panels), but arriving on 13 April 2015, 14:30–15:30 UTC between 2.3 and 4.9 km height. d) 96 h HYSPLIT backward trajectories arriving at Dushanbe at 18:00 UTC at 2.3, 3.6, and 4.9 km height.

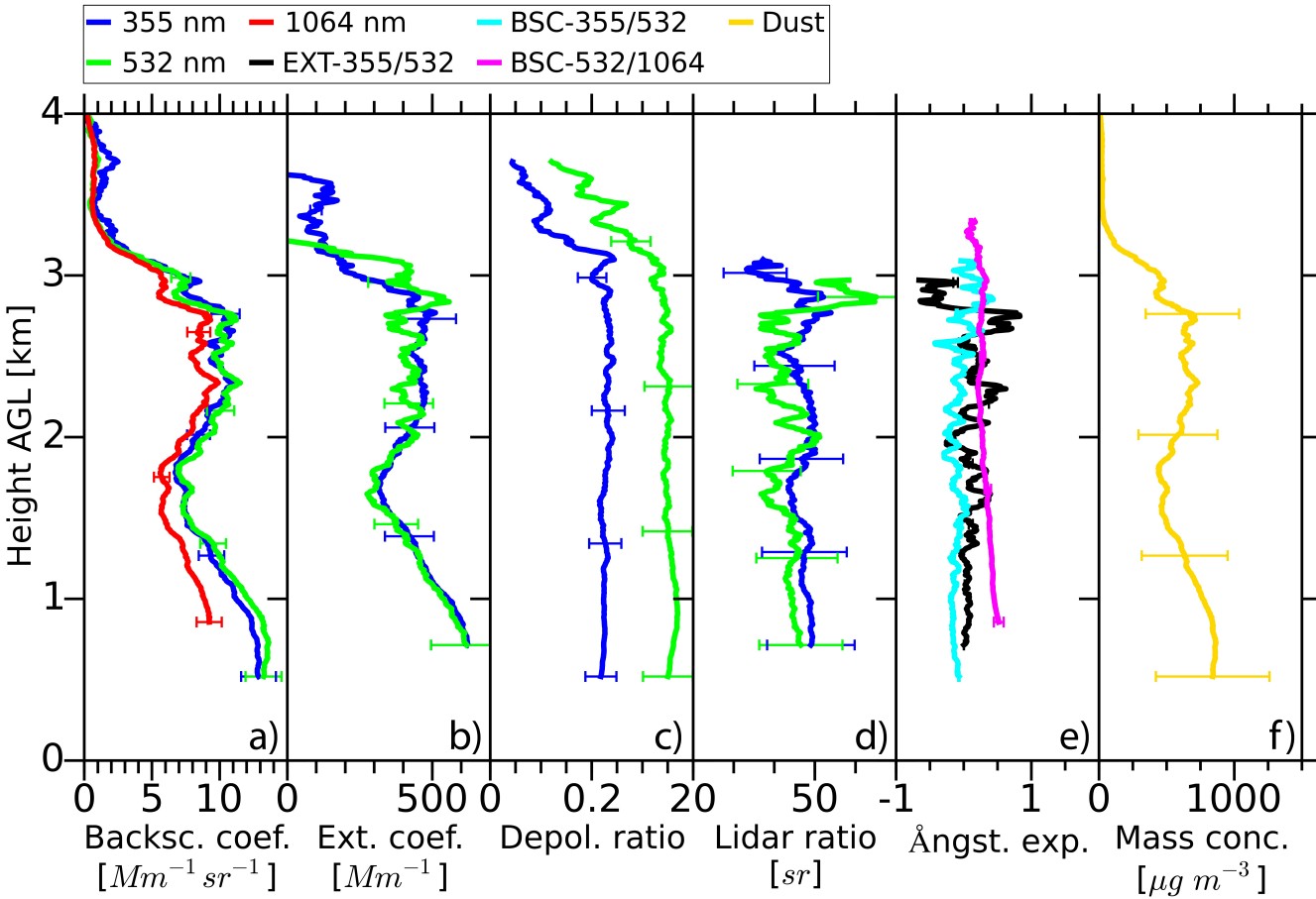

**Figure 6.** Case 2. An extreme dust layer with an AOT of 1.5 (at 532 nm wavelength) and a calculated dust mass concentration of 845 μg m$^{-3}$ is shown. The source regions of this dust are in Central Asia (Fig. 4f). a) – f) same as Fig. 5b. The averaged lidar profiles were measured on 8 August 2015, 22:20–23:57 UTC. Lidar signals are smoothed before calculation of the extinction-related (458 m) and backscatter-related (68 m) optical properties.

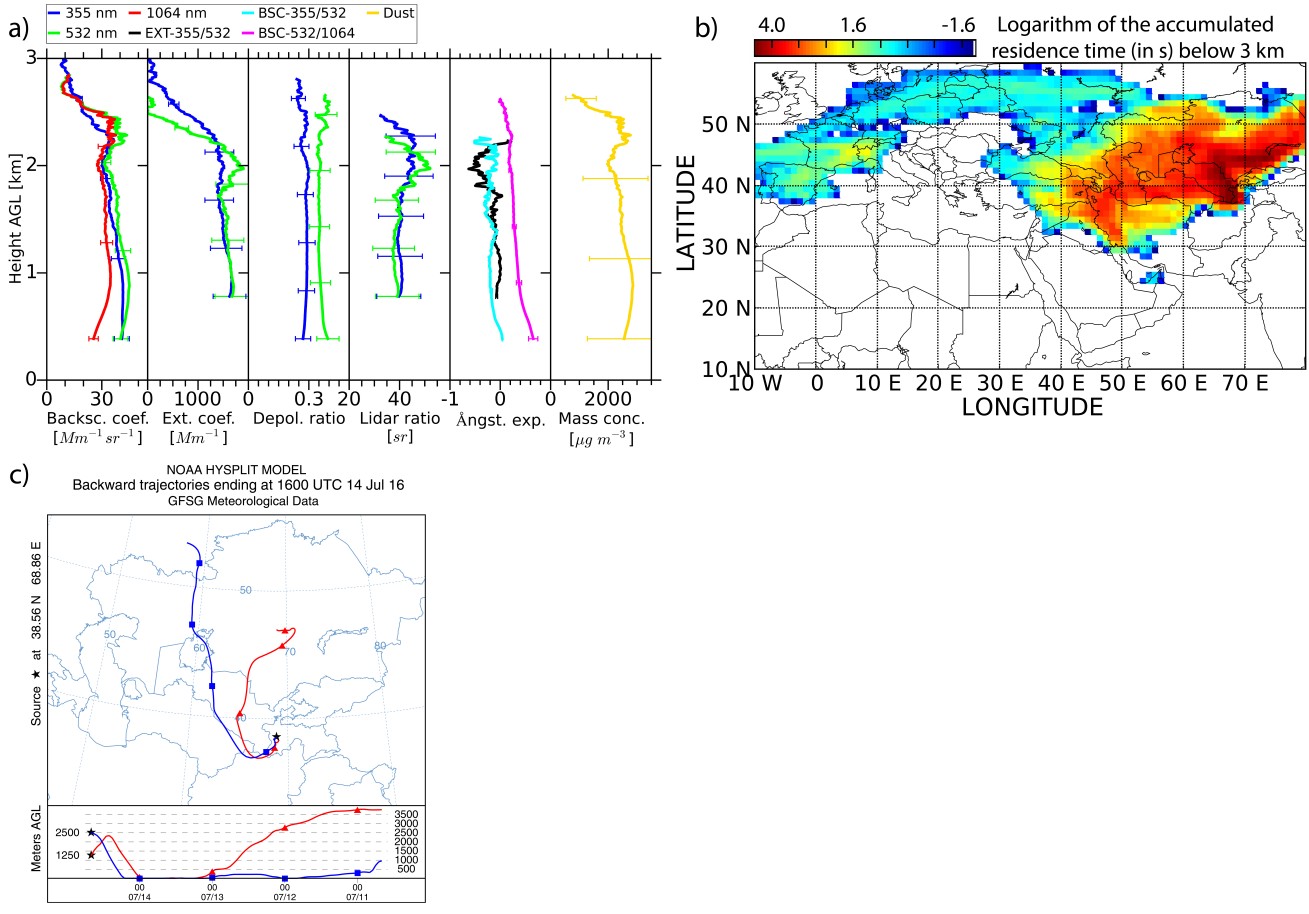

**Figure 7.** Case 3. The most extreme dust event with an AOT of above 4 (at 500 nm wavelength) and a calculated dust mass concentration of 2.8 mg m$^{-3}$ is shown. The dust source regions are in Central Asia. a) same as Fig. 5b. The averaged lidar profiles were measured on 14 July 2016, 16:00–22:00 UTC. Lidar signals are smoothed before calculation of the extinction-related (743 m) and backscatter-related (248 m) optical properties. b) 120 h FLEXPART backward trajectories same as in Fig. 4 (right panels), but arriving on 14 July 2016, 20:30–21:30 UTC between 0.5 and 2.6 km height. c) 96 h HYSPLIT backward trajectories arriving at Dushanbe at 16:00 UTC at 1.25, and 2.5 km height.

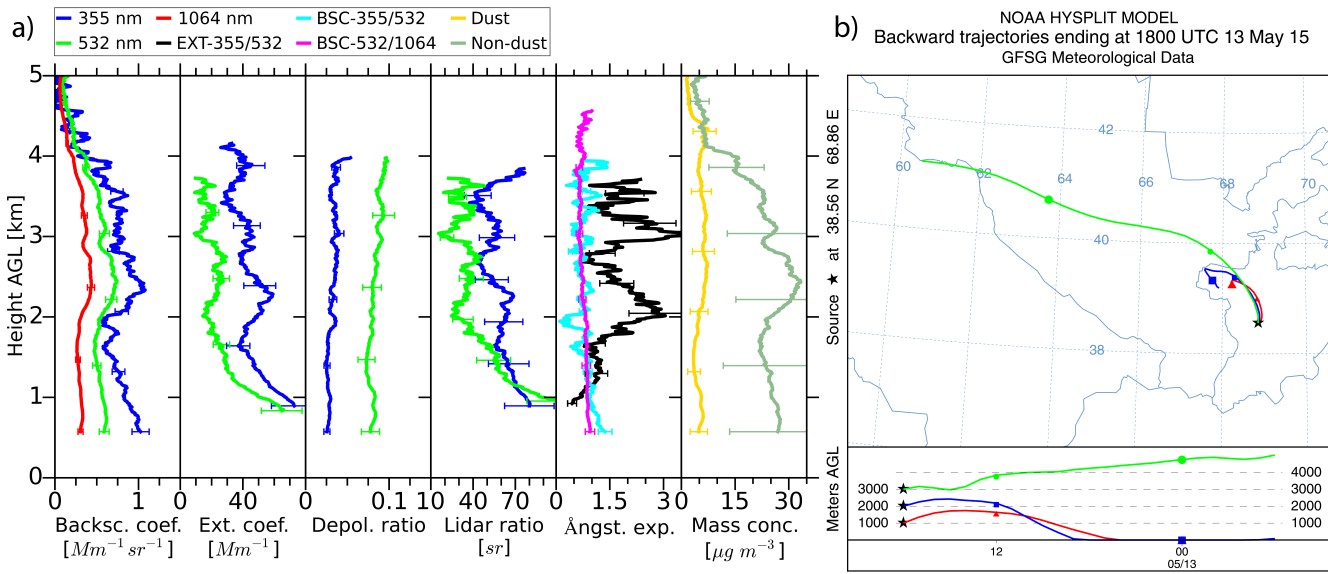

**Figure 8.** Case 4. A contrasting case with local pollution is shown. a) same as Fig. 5b. Averaged lidar profiles measured on 13 May 2015, 18:10–20:30 UTC. Lidar signals are smoothed before calculation of the extinction-related (788 m) and backscatter-related (98 m) optical properties. b) 48 h HYSPLIT backward trajectories arriving at Dushanbe at 18:00 UTC at 1, 2, and 3 km height.