# Peer review of "Long-term profiling of mineral dust and pollution aerosol with multiwavelength polarization/Raman lidar at the Central Asian site of Dushanbe, Tajikistan: Case studies"

_Atmospheric Chemistry and Physics, 2017_

## Referee Comment (RC1) · Anonymous Referee #1 · 16 Aug 2017

This is a first report on monitoring dust and atmospheric aerosol in central using a Raman lidar (POLLYxt), a sunphotometer, atmospheric models and satellite data. The authors very well reviewed previous works in this region, explained the geographical location of the investigation area, and very clearly reported four case studies during 18 months of measurements. The cases have been chosen for different types of origins to show the impact of influencing dust sources on the selected region. I believe this is a valuable manuscript that is quite fit for publication in ACP. Just there are some minor points that the author may consider them and the manuscript even now is in proper

form for publications. So there is no need that the manuscript would be reviewed again and after applying the following comments it will be well proper for publication.

1- In page 9 line 5, after the phrase "... maximum dust mass concentration of 196 $\mu$gm-3 ..." adding some reference to show that how the mass concentration can be extracted from the lidar signals is useful. 2- It is preferred to put labels (a, b, c, ...) on the sub-figures in figures 5, 7, and, 8 to address to the figure number and the label in the text, instead of addressing to their positions (top, left, ...) 3- In discussing case 3 on page 9 (14 July 2016), if a Hysplit back trajectory would be added to Fig. 7, this may help the reader to understand the case more clearly. 4- In page 10, line 31, inside the phrase "...second dust event the during..." the word "the" should be removed

Please also note the supplement to this comment:
https://www.atmos-chem-phys-discuss.net/acp-2017-559/acp-2017-559-RC1-supplement.pdf

---

## Referee Comment (RC2) · Anonymous Referee #3 · 16 Aug 2017

The long-term profiling of the dust and pollution aerosol has been conducted with multiwavelength polarization / Raman lidar at the Central Asia site of Dushanbe, Tajikistan as part of CADEX campaign. The site is located in the global dust belt, ranging from Sahara and Arabian deserts to the Taklimakan and Gobi deserts.

The present paper reported the case studies of CADEX campaign; AOT, Angstrom exponent, particle linear depolarization, lidar ratio and mass concentration for four cases. The sources and transport of the mineral dust were checked by back-trajectory and 3-D models. The present study will contribute a climatology of aerosol optical properties

in the global dust belt. It is accepted as ACP.

I have some comments as follows:

In page 6, line 10-14, the estimation of dust mass concentration by lidar is described. Did you validate this estimation with another method, such as a direct sampling?

In page 8, line 22-24, "Its base was very sharp ... into the boundary layer took place". In page 9, line 16-20, "Apparently, the layer is divided in two parts...by the diurnal cycle of the boundary layer." Information of the ABL, such as an inversion layer, a mixed layer etc. is helpful for these descriptions. Can you show data of the structure of ABL, such as radiosonde?

In page 10, line 7-19, the contribution of local pollution is indicated. Please specifically show a possible local source, and a constitution of the anthropogenic aerosols at the Central Asia site.

---

## Referee Comment (RC3) · Anonymous Referee #2 · 4 Sep 2017

The authors show case studies from lidar measurements of dust at Dushanbe in Central Asia. The paper is well structured and written and the results presented are very useful for global studies of dust and thus the paper should be considered for publication in ACP. There are few points that the authors should consider before publication.

Title: The authors mention "long term" and then "case studies". I believe that this is contradicting. There are no long-term measurement shown in the paper (18 months is not long-term), only case studies. It is not clear from the paper if this station will indeed be operational for many years to form a long-term record. I would suggest to remove

the term long-term from the title.

Introduction, Page 2, lines 20-28. Are there any studies on the mineralogical composition of dust in the area? Is the composition affected-related by the glacier shrinking?

Page 7, section 3.1 Overview. Why is it relevant to discuss in this section the dust events in the Mediterranean? I guess that the authors want just to use and cite a methodology for characterizing strong and extreme events. Since the paper does not make a global overview of strong dust events, is the comparison meaningful?

Section 4.3.and 4.2 A table would be very helpful to summarize the results for LR and depolarization ratios and eventually in this table the authors could include values from other studies with indication for the origin and the purity of dust or mixture status. Do the author finally claim that they observed also pure dust at Dushanbe? Is it possible then to separate for the other cases the dust component?

Does the agreement with the LRs measured for West-Asian dust indicates similar mineralogical composition? Please comment on this.

Page 11, line 24, check the wavelength it is mentioned as 335nm.

---

## Author Comment (AC1) · 10 Oct 2017

We thank the reviewers for the careful reading of this contribution. The questions and useful suggestions helped us a lot to make the statements of this article clearer. The letter of reply (supplement) contains comments to all three referees and the changes.

Please also note the supplement to this comment: https://www.atmos-chem-phys-discuss.net/acp-2017-559/acp-2017-559-AC1-supplement.pdf

[Figure]

[Figure]

**Supplement:**

We thank the reviewers for the careful reading of this contribution. The questions and useful suggestions helped us a lot to make the statements of this article clearer. The textual changes are marked with red in the revised manuscript, which is attached at the end of this supplement.

**Comments to Referee 1**

*In page 9 line 5, after the phrase "... maximum dust mass concentration of 19 $\mu gm^{-3}$ ..." adding some reference to show that how the mass concentration can be extracted from the lidar signals is useful.*

**A sentence with a reference is added in the instruments section (page 6, line 17). Please see also the comment to Referee 3 on the POLIPHON method.**

*It is preferred to put labels (a, b, c,...) on the sub-figures in figures 5, 7, and, 8 to address to the figure number and the label in the text, instead of addressing to their positions (top, left,...)*

**This is corrected, the sub-figures have now alphabetical labels. The sub-subfigures of the vertical profiles are now without alphabetical (sub-)labeling.**

*In discussing case 3 on page 9 (14 July 2016), if a Hysplit back trajectory would be added to Fig. 7, this may help the reader to understand the case more clearly.*

**A Hysplit backward trajectory is now added to the figure (Fig 7, page 33) and shortly discussed in the text (page 9, line 31).**

*In page 10, line 31, inside the phrase "...second dust event the during..." the word "the" should be removed.*

**The typing error is now corrected (page 11, line 1).**

**Comments to Referee 2**

*Title: The authors mention "long term" and then "case studies". I believe that this is contradicting. There are no long-term measurement shown in the paper (18 months is not long-term), only case studies. It is not clear from the paper if this station will indeed be operational for many years to form a long-term record. I would suggest to remove the term long-term from the title.*

**Long-term is a relative word. We have measured for almost 500 days in Dushanbe, what is significantly more than other ground-based lidar field campaigns (e.g. SAMUM-1, -2, SALTRACE, SHADOW) did before at other rather remote places on earth. Furthermore, with this 18-months period we covered two spring and two summer seasons. It is made clear, that this study is a "part 1" study, where typical case studies are discussed. In a follow up "part**

2" study the statistical analysis of the whole measurement period will follow. The last sentence of the manuscript has been changed (page 13, line 25) to indicate more clearly that the lidar station in Tajikistan will become a permanent station soon, probably starting from 2019, which allows longer multi-year observations, maybe even to monitor longer-term trends.

*Introduction, Page 2, lines 20-28. Are there any studies on the mineralogical composition of dust in the area? Is the composition affected-related by the glacier shrinking?*
To comprehensively discuss the composition of the mineral dust in Central Asia or even asses the composition's effect on the glacier shrinking is far beyond the scope of this study. We can provide height-resolved measurements of optical properties of the atmospheric mineral dust, which could be used by modelers for example for radiative transfer calculations or estimations of dust settling on glacier surfaces. There is a sentence added mentioning a couple of mineralogical studies in the area (page 3, line 13). Please see further comment on mineralogy/lidar ratio below.

*Page 7, section 3.1 Overview. Why is it relevant to discuss in this section the dustevents in the Mediterranean? I guess that the authors want just to use and cite a methodology for characterizing strong and extreme events. Since the paper does not make a global overview of strong dust events, is the comparison meaningful?*
Yes, we want to use this methodology to classify our measurements and compare them to another region (Mediterranean area) close to various dust source regions. The sentence (page 7, line 23) citing the long-term mean aerosol optical thickness at 500 nm in the Eastern Mediterranean (Georgoulias et al., 2016) indeed has to be sufficient for that comparison. The last two sentences on page 7 are therefore deleted.

*Section 4.3.and 4.2 A table would be very helpful to summarize the results for LR and depolarization ratios and eventually in this table the authors could include values from other studies with indication for the origin and the purity of dust or mixture status.*
We decided to have a table comparing the four discussed example cases. In the discussion section the measured values of the particle linear depolarization ratio and the lidar ratio are compared to measurements of Saharan, Middle Eastern, and especially to Asian dust. The range of the reported depolarization values in literature is rather large. Not every mentioned study makes statements on the purity of the measured dust or even on mixing ratios determined for example by Burton et al. (2014). On the one hand, the values (0.27-0.35, mean 0.31, at 532 nm) of Saharan dust are reported to be from pure dust (Freudenthaler et al., 2009) and Groß et al. (2015a) uses a range of 0.23-0.33 at 532 nm to identify pure Saharan dust (together with values for the lidar ratio). The lower edges of those ranges could be already classified as slightly polluted/mixed dust (Groß et al., 2015b, 2013) (see also next comment below). On the other hand,

the cited studies for Asian dust (e.g. Dieudonné et al., 2015; Kai et al., 2008; Chen et al., 2013; Chen and Sverdlik, 2007) show even a higher variability reaching from 0.09 to 0.33 at 532 nm, probably also describing polluted/mixed dust episodes. To indicate this fact, a sentence (page 12, line 25) was changed to: "The measured depolarization ratios for the presented dust cases (Cases 1,2,3) are mostly higher than these literature values for dust measured in or close to Central Asia. This suggests that those studies might have described also observations of polluted/mixed dust."

*Do the author finally claim that they observed also pure dust at Dushanbe? Is it possible then to separate for the other cases the dust component?*
We estimate the purity of the dust based on the measured optical properties, namely the particle linear depolarization ratio. For POLIPHON method we used particle linear depolarization threshold values (at 532 nm wavelength) of 0.05 for non-dust and 0.31 for (pure) dust. Based on this discrimination we separate the corresponding backscatter profiles into their dust and non-dust components. These components then get converted to extinction (with specific lidar ratios) and then to mass (with specific extinction-to-mass conversion factors). So each dust/non-dust mass concentration profile basically is a separation into dust/non-dust. Concerning the four example cases: Case 1 has depolarization ratio of about 0.31 up to about 2.3–3.9 km height and even up to 0.34 in its lowest parts (this is corrected on page 9, line 2). On the upper part of the dust layer the depolarization ratio is a bit lower, on average 0.30 (3.9–4.9 km). The average depolarization ratio over the whole extent of the dust layer is 0.31 (2.3–4.9 km). So the POLIPHON method would only yield a vanishingly low non-dust component up to about 3.9 km, and between 3.9 and 4.9 km about a 10 times lower non-dust than dust mass concentration. To make this more clear, one sentence is added (page 9, line 7): "The non-dust components in this layer are negligible, except in the upper parts of the layer, where the particle linear depolarization ratio partly drops below 0.31 at 532 nm wavelength". The cases 2 and 3 are separated into dust only, since they have depolarization ratios of constantly above 0.31. So the non-dust mass concentration is not plotted for these three cases. In case 4 however, the aerosol layer (13 May 2015) has a lower depolarization ratio, below 0.1 at 532 nm wavelength, so both dust and non-dust mass concentrations profiles are shown.

*Does the agreement with the LRs measured for West-Asian dust indicates similar mineralogical composition? Please comment on this.*
From our lidar measurements we cannot judge which mineralogical composition the measured dust has. Results from the aerosol collected in situ at ground with the high volume sampler are not fully analyzed yet and therefore cannot be subject of this study. Furthermore, the comparison of ground-based in situ measurements of chemical and mineralogical composition with the optical

properties measured by lidar is challenging, especially in the case of lofted dust layers which could possibly not be probed by the in situ instruments.

The mineralogical differences as well as their impact on optical/radiative properties between Asian and African dust have been discussed in several studies (e.g. Fitzgerald et al., 2015; Formenti et al., 2011; Su and Toon, 2011; Di Biagio et al., 2014).

The reason for the lower lidar ratios of Arabian/West-Asian dust compared to Saharan dust is not known up to now. There is a theory by Schuster et al. (2012). This article is cited in the discussion section about the lidar ratios, page 11, line 27. Schuster et al. (2012) use AERONET data to retrieve/invert pure dust lidar ratios (Dubovik et al., 2006). They find lower values for Arabian dust than for Saharan dust and also a strong anti-correlation of the retrieved/inverted lidar ratio with the real refractive index. From that finding they hypothesize that the illite content determines to a certain degree the lidar ratio of dust, because illite has a low real refractive index. They then compare literature values (Kalderon-Asael et al., 2009; Ganor, 1991; Caquineau et al., 2002) for illite content in soil at the potential dust source regions and found a spatial agreement to their hypothesis. In contrast, Lafon et al. (2006) state that "Illite is an ubiquitous specie, and it has been found to be the major clay in mineral dust aerosols originating from the Sahara [Chester et al., 1977] and from the Asian continent [Merrill et al., 1994]". Singer (1988) even finds that "In some desert loess deposits, notably those of Central Asia, illite dominates the clay fraction, in others, such as in the Near East and North Africa, illite is a minor component."

Fitzgerald et al. (2015) state that "Saharan dust is rich in aluminosilicates and clays, and Asian dust is calcite-rich (Formenti et al., 2011; Hatch and Grassian, 2008)". Some studies indeed report atmospheric dust in Central Asia having a calcareous character and additionally high contents of soluble salts (Miller-Schulze et al., 2011; Schettler et al., 2014; Groll et al., 2013; Andronova et al., 1993). We added the following sentence (page 3, line 13): "Chemical analysis of the collected dust (Gomes and Gillette, 1993) showed a low iron content for Central Asian dust. Also, a calcareous character and high contents of soluble salts were observed (Miller-Schulze et al., 2011; Schettler et al., 2014; Groll et al., 2013; Andronova et al., 1993). In general, however, a very high variability (Andronova et al., 1993) lead to considerably different results depending on sampling location".

Another approach would be to look at the iron(oxide) content, which has an effect on the optical properties of dust (e.g. Sokolik and Toon, 1999; Derimian et al., 2008), but it depends also on the kind of oxide (goethite, hematite) and then again on the clay component (illite, kaolinite) in the aggregate (Lafon et al., 2006). The study of Gomes and Gillette (1993) during the Soviet-American campaign in Tajikistan in 1989 states that "The Sahelian regions present abundant ferralitic soils (Millat, 1964; Sys, 1967) whereas the Central

Asian regions do not. Therefore the scarcity of the Fe in the dust deposited in Tajikistan seems to constitute a good signature for that region of Central Asia" (This is cited in the manuscript on page 3, line 14). Middle Eastern dust on the other hand is reported to have higher iron content than North African dust (Jin et al., 2016; Journet et al., 2014; Nickovic et al., 2012), what would compromise the - probably also too simple - hypothesis that the lower lidar ratios of West and Central Asian dust could be determined by a lower iron content.

So, we do not want to speculate about a possible link between mineralogy and lidar ratio, since we cannot judge it by our lidar measurements alone, as above mentioned. But we consider lidar field campaigns, especially continuous, long-term measurements, together with in situ observations and laboratory studies as essential to get a better understanding of optical, radiative, microphysical, and mineralogical properties of (Central Asian) dust.

*Page 11, line 24, check the wavelength it is mentioned as 335 nm.*
The sentence is corrected and is now stating the correct wavelength (page 11, line 25).

**Comments to Referee 3**

*In page 6, line 10-14, the estimation of dust mass concentration by lidar is described. Did you validate this estimation with another method, such as a direct sampling?*
No, for the Tajikistan site it was not possible to validate this estimation with in situ sampling.
The POLIPHON method is based on relationships between particle backscatter, extinction, surface area, and volume concentrations for different aerosol types (Ansmann et al., 2012) that are discussed in Barnaba and Gobbi (2001, 2002). These relationships were validated by Gobbi et al. (2003).
The POLIPHON method has been compared for example to LIRIC (Lidar-Radiometer Inversion Code) inversions (Wagner et al., 2013), where acceptable agreement was found, and the LIRIC inversion was compared to in situ airborne measurements and good agreement was found (Kokkalis et al., 2017). Bravo-Aranda et al. (2015) compared the POLIPHON method directly to in situ aircraft measurements and find less than 30% relative difference (within the error margin of 30–50%). A sentence mentioning this is added in the instruments section (page 6, line 17).
As stated in the instruments section, a Grimm particle counter (note: again an optical measurement device) and a Digitel high volume sampler were operated on ground in Dushanbe. The gathered data could be used to try to validate the POLIPHON calculations (at least at ground/PBL) for Tajikistan in a follow up study.

*In page 8, line 22-24, "Its base was very sharp ... into the boundary layer took place". In page 9, line 16-20, "Apparently, the layer is divided in two parts...by the diurnal cycle of the boundary layer." Information of the ABL, such as an inversion layer, a mixed layer etc. is helpful for these descriptions. Can you show data of the structure of ABL, such as radiosonde?*

There were no radiosonde launches in Tajikistan since several years. The GDAS profiles for that day (Fig. C1) show no variability up to 3 km height. The mentioned statements are based on judging the development of the planetary boundary layer by considering the temporal development of the range-corrected signal at 1064 nm. In the case of the 8 August 2015, the intensive optical properties (particle linear depolarization and lidar ratio, Ångström exponent) in the twofold structured layer are almost identical. This is used to justify the averaging of that values in a height range of both of the apparent layers. To clarify this, the text is changed mentioning the temporal development of the range-corrected signal at 1064 nm (page 9, line 21). In Case 1, 13 April 2015, the text was changed to "the lowermost 2 km" (page 8, line 24).

[Figure]

Figure C1: GDAS relative humidity (left) and temperature profiles (right) on 8 August 2015 at the coordinates 39° N, 69° E.

*In page 10, line 7-19, the contribution of local pollution is indicated. Please specifically show a possible local source, and a constitution of the anthropogenic aerosols at the Central Asia site.*

There are a couple of possible sources which could be mentioned, for example the traffic/fuel combustion (older cars), factories, especially a cement factory and a coal power plant in Dushanbe. It is made more clear that the study of

Hansen et al. (1993), cited in the introduction section, was using filter samples from the city of Dushanbe (page 3, line 17), a city "with copious fuel combustion emissions visible and a distinct air pollution problem" (Hansen et al., 1993). We were measuring optical properties and classify the aerosol based on those properties, in this case specifically the particle linear depolarization ratio. We cannot provide the exact constitution of that aerosol.

**References used in supplement**

Andronova, A. V., Gomes, L., Smirnov, V. V., Ivanov, A. V., and Shukurova, L. M.: Physico-chemical characteristics of dust aerosols deposited during the Soviet-American experiment (Tadzhikistan, 1989), Atmos. Environ., 27, 2487–2493, doi: 10.1016/0960-1686(93)90020-Y, 1993.

Ansmann, A., Seifert, P., Tesche, M., and Wandinger, U.: Profiling of fine and coarse particle mass: case studies of Saharan dust and Eyjafjallajökull/Grimsvötn volcanic plumes, Atmos. Chem. Phys., 12, 9399–9415, doi:10.5194/acp-12-9399-2012, 2012.

Barnaba, F. and Gobbi, G. P.: Lidar estimation of tropospheric aerosol extinction, surface area and volume: Maritime and desert-dust cases, J. Geophys. Res., 106, 3005–3018, doi:10.1029/2000JD900492, 2001.

Barnaba, F. and Gobbi, G. P.: Correction to "Lidar estimation of tropospheric aerosol extinction, surface area and volume: Maritime and desert-dust cases", J. Geophys. Res., 107, 4180, doi:10.1029/2002JD002340, 2002.

Bravo-Aranda, J. A., Titos, G., Granados-Muñoz, M. J., Guerrero-Rascado, J. L., Navas-Guzmán, F., Valenzuela, A., Lyamani, H., Olmo, F. J., Andrey, J., and Alados-Arboledas, L.: Study of mineral dust entrainment in the planetary boundary layer by lidar depolarisation technique, Tellus B, 67, 26 180, doi:10.3402/tellusb.v67.26180, 2015.

Burton, S. P., Vaughan, M. A., Ferrare, R. A., and Hostetler, C. A.: Separating mixtures of aerosol types in airborne High Spectral Resolution Lidar data, Atmos. Meas. Tech., 7, 419–436, doi:10.5194/amt-7-419-2014, 2014.

Caquineau, S., Gaudichet, A., Gomes, L., and Legrand, M.: Mineralogy of Saharan dust transported over Northwestern tropical Atlantic Ocean in relation to source regions, J. Geophys. Res. Atmos., 107, doi:10.1029/2000JD000247, 2002.

Chen, B. and Sverdlik, L.: Optical and microphysical characteristics of aerosol structures in Central Asia, in: International Conf. on Lasers, Applications, and Technologies 2007: Environmental Monitoring and Ecological Applications; Optical Sensors in Bio, Chemical, and Engineering Technologies; and Femtosecond Laser Pulse Filamentation, vol. 6733, doi:10.1117/12.753117, 2007.

Chen, B. B., Sverdlik, L. G., Imashev, S. A., Solomon, P. A., Lantz, J., Schauer, J. J., Shafer, M. M., Artamonova, M. S., and Carmichael, G.: Lidar Measurements of the Vertical Distribution of Aerosol Optical and Physical Properties over Central Asia, Air Qual. Atmos. Health, p. 385–396, doi:10.1007/s11869-012-0192-5, 2013.

Derimian, Y., Karnieli, A., Kaufman, Y. J., Andreae, M. O., Andreae, T. W., Dubovik, O., Maenhaut, W., and Koren, I.: The role of iron and black carbon in aerosol light absorption, Atmos. Chem. and Phys., 8, 3623–3637, doi:10.5194/acp-8-3623-2008, 2008.

Di Biagio, C., Formenti, P., Styler, S. A., Pangui, E., and Doussin, J.-F.: Laboratory chambermeasurements of the longwave extinction spectra and complex refractive indices of African and Asian mineral dusts, Geophys. Res. Lett., 41, 6289–6297, doi:10.1002/2014GL060213, 2014.

Dieudonné, E., Chazette, P., Marnas, F., Totems, J., and Shang, X.: Lidar profiling of aerosol optical properties from Paris to Lake Baikal (Siberia), Atmos. Chem. Phys., 15, 5007–5026, doi:10.5194/acp-15-5007-2015, 2015.

Dubovik, O., Sinyuk, A., Lapyonok, T., Holben, B., Mishchenko, M., Yang, P., Eck, T. F., Volten, H., Muñoz, O., Veihelmann, B., van der Zande, W. J., Leon, J.-F., Sorokin, M., and Slutsker, I.: Application of spheroid models to account for aerosol particle nonsphericity in remote sensing of desert dust, J. Atmos. Sci., 59, 590–608, doi:10.1175/1520-0469(2002)0592.0.CO;2, 2006.

Fitzgerald, E., Ault, A. P., Zauscher, M. D., Mayol-Bracero, O. L., and Prather, K. A.: Comparison of the mixing state of long-range transported Asian and African mineral dust, Atmos. Env., 115, 19–25, doi:10.1016/j.atmosenv.2015.04.031, 2015.

Formenti, P., Schütz, L., Balkanski, Y., Desboeufs, K., Ebert, M., Kandler, K., Petzold, A., Scheuvens, D., Weinbruch, S., and Zhang, D.: Recent progress in understanding physical and chemical properties of African and Asian mineral dust, Atmos. Chem. Phys., 11, 8231–8256, doi:10.5194/acp-11-8231-2011, 2011.

Freudenthaler, V., Esselborn, M., Wiegner, M., Heese, B., Tesche, M., Ansmann, A., Müller, D., Althausen, D., Wirth, M., Fix, A., Ehret, G., Knippertz, P., Toledano, C., Gasteiger, J., Garhammer, M., and Seefeldner, M.: Depolarization ratio profiling at several wavelengths in pure Saharan dust during SAMUM 2006, Tellus B, 61, 165–179, doi:10.1111/j.1600-0889.2008.00396.x, 2009.

Ganor, E.: The composition of clay minerals transported to Israel as indicators of Saharan dust emission, Atmos. Environ., 25, 2657–2664, doi:10.1016/0960-1686(91)90195-D, 1991.

Georgoulias, A. K., Alexandri, G., Kourtidis, K. A., Lelieveld, J., Zanis, P., Pöschl, U., Levy, R., Amiridis, V., Marinou, E., and Tsikerdekis, A.: Spatiotemporal variability and contribution of different aerosol types to the aerosol optical depth over

the Eastern Mediterranean, Atmos. Chemis. Phys., 16, 13 853–13 884, doi:10.5194/ acp-16-13853-2016, 2016.

Gobbi, G. P., Barnaba, F., Van Dingenen, R., Putaud, J. P., Mircea, M., and Facchini, M. C.: Lidar and in situ observations of continental and Saharan aerosol: closure analysis of particles optical and physical properties, Atmos. Chem. and Phys., 3, 2161–2172, doi: 10.5194/acp-3-2161-2003, 2003.

Gomes, L. and Gillette, D. A.: A comparison of characteristics of aerosol from dust storms in Central Asia with soil-derived dust from other regions, Atmos. Environ., 27, 2539–2544, doi:10.1016/0960-1686(93)90027-V, 1993.

Groll, M., Opp, C., and Aslanov, I.: Spatial and temporal distribution of the dust deposition in Central Asia – results from a long term monitoring program, Aeolian Res., 9, 49–62, doi:10.1016/j.aeolia.2012.08.002, 2013.

Groß, S., Esselborn, M., Weinzierl, B., Wirth, M., Fix, A., and Petzold, A.: Aerosol classification by airborne high spectral resolution lidar observations, Atmos. Chem. Phys., 13, 2487–2505, doi:10.5194/acp-13-2487-2013, 2013.

Groß, S., Freudenthaler, V., Schepanski, K., Toledano, C., Schäfler, A., Ansmann, A., and Weinzierl, B.: Optical properties of long-range transported Saharan dust over Barbados as measured by dual-wavelength depolarization Raman lidar measurements, Atmos. Chem. Phys., 15, 11 067–11 080, doi:10.5194/acp-15-11067-2015, 2015a.

Groß, S., Freudenthaler, V., Wirth, M., and Weinzierl, B.: Towards an aerosol classification scheme for future EarthCARE lidar observations and implications for research needs, Atmos. Sci. Let., 16, 77–82, doi:10.1002/asl2.524, 2015b.

Hansen, A. D. A., Kapustin, V. A., Kopeikin, V. M., Gillette, D. A., and Bodhaine, B. A.: Optical absorption by aerosol black carbon and dust in a desert region of Central Asia, Atmos. Environ., 27, 2527–2531, doi:10.1016/0960-1686(93)90025-T, 1993.

Jin, Q., Yang, Z.-L., and Wei, J.: High sensitivity of Indian summer monsoon to Middle East dust absorptive properties, Sci. Rep., 6, 30 690, doi:10.1038/srep30690, 2016.

Journet, E., Balkanski, Y., and Harrison, S. P.: A new data set of soil mineralogy for dust-cycle modeling, Atmos. Chem. and Phys., 14, 3801–3816, doi:10.5194/acp-14-3801-2014, 2014.

Kai, K., Nagata, Y., Tsunematsu, N., Matsumura, T., Kim, H.-S., Matsumoto, T., Hu, S., Zhou H., Abo, M., and Nagai, T.: The structure of the dust layer over the Taklimakan Desert during the dust storm in April 2002 as observed using a depolarization lidar, J. Meteor. Soc. Japan, 86, 1–16, doi:10.2151/jmsj.86.1, 2008.

Kalderon-Asael, B., Erel, Y., Sandler, A., and Dayan, U.: Mineralogical and chemical characterization of suspended atmospheric particles over the East Mediterranean based on synoptic-scale circulation patterns, Atmos. Environ., 43, 3963–3970, doi:10.1016/j.atmosenv.2009.03.057, 2009.

Kokkalis, P., Amiridis, V., Allan, J. D., Papayannis, A., Solomos, S., Binietoglou, I., Bougiatioti, A., Tsekeri, A., Nenes, A., Rosenberg, P. D., Marenco, F., Marinou, E., Vasilescu, J., Nicolae, D., Coe, H., Bacak, A., and Chaikovsky, A.: Validation of LIRIC aerosol concentration retrievals using airborne measurements during a biomass burning episode over Athens, Atmos. Res., 183, 255–267, doi:10.1016/j.atmosres.2016.09.007, 2017.

Lafon, S., Sokolik, I. N., Rajot, J. L., Caquineau, S., and Gaudichet, A.: Characterization of iron oxides in mineral dust aerosols: Implications for light absorption, J. Geophys. Res. Atmos, 111, D21 207, doi:10.1029/2005JD007016, 2006.

Miller-Schulze, J. P., , Shafer, M. M., Schauer, J. J., A., S. P., Lantz, J., Artamonova, M., Chen, B. Imashev, S., Sverdlik, L., Carmichael, G. R., and Deminter, J. T.: Characteristics of fine particle carbonaceous aerosol at two remote sites in Central Asia, Atmos. Environ., 45, 695–6964, doi:10.1016/j.atmosenv.2011.09.026, 2011.

Nickovic, S., Vukovic, A., Vujadinovic, M., Djurdjevic, V., and Pejanovic, G.: Technical Note: High-resolution mineralogical database of dust-productive soils for atmospheric dust modeling, Atmos. Chem. and Phys., 12, 845–855, doi:10.5194/acp-12-845-2012, 2012.

Schettler, G., Shabunin, A., Kemnitz, H., Knoeller, K., Imashev, S., Rybin, A., and Wetzel, H.-U.: Seasonal and diurnal variations in dust characteristics on the northern slopes of the Tien Shan – Grain-size, mineralogy, chemical signatures and isotope composition of attached nitrate, J. Asian Earth Sci., 88, 257–276, doi:10.1016/j.jseaes.2014.03.019, 2014.

Schuster, G. L., Vaughan, M., MacDonnell, D., Su, W., Winker, D., Dubovik, O., Lapyonok, T., and Trepte, C.: Comparison of CALIPSO aerosol optical depth retrievals to AERONET measurements, and a climatology for the lidar ratio of dust, Atmos. Chem. Phys., 12, 7431–7452, doi:10.5194/acp-12-7431-2012, 2012.

Singer, A.: Illite in aridic soils, desert dusts and desert loess, Sediment. Geol., 59, 251–259, doi:10.1016/0037-0738(88)90079-6, 1988.

Sokolik, I. N. and Toon, O. B.: Incorporation of mineralogical composition into models of the radiative properties of mineral aerosol from UV to IR wavelengths, J. Geophys. Res. Atmos., 104, 9423–9444, doi:10.1029/1998JD200048, 1999.

Su, L. and Toon, O. B.: Saharan and Asian dust: similarities and differences determined by CALIPSO, AERONET, and a coupled climate-aerosol microphysical model, Atmos. Chem. Phys., 11, 3263–3280, doi:10.5194/acp-11-3263-2011, 2011.

Wagner, J., Ansmann, A., Wandinger, U., Seifert, P., Schwarz, A., Tesche, M., Chaikovsky, A., and Dubovik, O.: Evaluation of the Lidar/Radiometer Inversion Code (LIRIC) to determine microphysical properties of volcanic and desert dust, Atmos. Meas. Tech., 6, 1707–1724, doi:10.5194/amt-6-1707-2013, 2013.

[revised manuscript text omitted]